



# Propagation of gravity waves and its effects on pseudomomentum flux in a sudden stratospheric warming event

In-Sun Song[1], Changsup Lee[1], Hye-Yeong Chun[2], Jeong-Han Kim[1], Geonhwa Jee[1],
Byeong-Gwon Song[1], and Julio T. Bacmeister[3]

[1]Korea Polar Research Institute, Incheon, Korea
[2]Department of Atmospheric Sciences, Yonsei University, Seoul, Korea
[3]National Center for Atmospheric Research, Boulder, CO, USA

**Correspondence:** In-Sun Song (isong@kopri.re.kr)

**Abstract.** Effects of realistic propagation of gravity waves (GWs) on distribution of GW pseudomomentum fluxes ($F_p$s) are explored using a global ray-tracing model for the 2009 sudden stratospheric warming (SSW). Four-dimensional (4D) ($x$–$z$, $t$) and two-dimensional (2D) ($z$, $t$) results are compared for various parameterized $F_p$s. In ray-tracing equations, refraction due to horizontal wind shear and curvature effects are found important and comparable to one another in magnitude. In the 4D, westward $F_p$s are enhanced in the upper troposphere and northern stratosphere, due to refraction and curvature effects around fluctuating jet flows associated with large-scale waves. In the northern polar upper mesosphere and lower thermosphere, eastward $F_p$s are increased in the 4D. GWs are found to propagate more to the upper atmosphere in the 4D, since horizontal propagation and change in wavenumbers due to refraction and curvature effects can make it more possible that GWs elude critical-level filtering and saturation in the lower atmosphere. GW focusing and ray-tube effects have some impacts on changes in $F_p$s. Focusing effects occur around jet cores, and ray-tube effects appear where the polar stratospheric jets vary substantially in space and time. Increase in the $F_p$s in the northern upper stratosphere and the lower thermosphere begins from the early stage of the SSW evolution, and it is present even after the onset in the 4D. Significantly enhanced $F_p$s in the northern stratosphere are likely related to GWs with small intrinsic group velocity (wave capture), and they would change nonlocally nearby large-scale vortex structure without changing substantially local mean flows.

## 1 Introduction

Atmospheric gravity waves (GWs) may have profound impacts in momentum and energy budgets of global circulations in the middle and upper atmospheres. GW pseudomomentum fluxes can induce large-scale momentum forcing, which can substantially change ambient winds, when either transience related to unsteady propagation or dissipation due to breaking or damping occurs (e.g., Fritts and Alexander, 2003; Bühler, 2014).

GWs may also affect global thermal structure through adiabatic vertical motions and heat deposition. GW momentum forcing induces the meridional and vertical mass circulations that can contribute to temperature structure related to the Brewer-Dobson circulation in the stratosphere (e.g., Rosenlof and Holton, 1993; Chun et al., 2011; Seviour et al., 2012) and can reverse the radiatively-driven latitudinal temperature gradient across the two poles in the upper mesosphere (e.g., Kim et al., 2003; Smith



et al., 2012). Irreversible heat and heat fluxes can occur when GW momentum forcing is induced (e.g., Becker and Schmitz, 2002; Medvedev and Klaassen, 2003; Yiğit and Medvedev, 2009), and they produce GW-induced heat deposition.

In general, excitation of GWs is unsteady, and GWs propagate at finite group velocities in the form of localized packets or wavetrains. Hence, studies on propagation of GW packets in slowly-varying large-scale flows have been carried out using

ray-tracing modeling based on the spatial ray theory (e.g. Dunkerton, 1984; Marks and Eckermann, 1995). Hasha et al. (2008) extended the ray theory to the spherical geometry. Ribstein et al. (2015) presented more complete formulations in which the magnitude of the three-dimensional (3D) wavenumber vector is invariant with respect to the earth's curvature under deep atmosphere approximation.

In GW parameterizations (GWPs) for global climate and numerical prediction models, however, propagation of GWs has

been dealt with under simplifying assumptions that steady GWs propagate instantaneously only in the vertical direction from tropospheric sources to model top. To consider horizontal and time propagation of GWs, Song and Chun (2008) developed a ray-based GWP for convective GWs for use in Whole Atmosphere Community Climate Model (WACCM). Senf and Achatz (2011) employed a ray-based method to compute GW-induced forcing in the spatiotemporally varying large-scale flow associated with the thermal tides in the mesosphere and lower thermosphere (MLT), and they discussed validity of simplifying

assumptions in conventional GWPs. Kalisch et al. (2014) showed importance of momentum forcing due to poleward propagating GWs using a ray-tracing model, and they discussed implementation of effects of poleward propagation in global models. Amemiya and Sato (2016) presented a quasi-columnar way of implementing a ray-based GWP in global models, ignoring time propagation of GWs. Yet, it is not clear how ray-based GWPs can be formulated in a way consistent with theories on interaction between GW packets and slowly-varying mean flows. Besides, implementation of ray GWPs in models is not straightforward

since it requires overcoming limitations of conventional modeling frameworks where all parameterizations are columnar, and subgrid-scale processes across timesteps (e.g., time-propagating GWs) are ignored.

There have been studies in idealized settings to understand effects of horizontal and transient propagation of GWs on interaction between GWs and slowly-varying mean flows. Bühler and McIntyre (2003) presented a theory on wave-mean interaction associated with horizontal refraction of GWs. Bühler and McIntyre (2005) demonstrated a new type of interaction (wave

capture) between GWs and horizontally varying vortices using conservation law for the sum of GW pseudomomentum and impulse for GW packets in slowly-varying mean flows. Eckermann et al. (2015) investigated horizontal spreading of orographic GW rays and showed that horizontal spreading can be as important as refraction of vertical wavenumbers in the wave-mean interaction.

Dunkerton (1981) demonstrated analytically and numerically that transient waves with finite vertical group velocities can

induce spontaneous mean-flow responses such as descent of mean shear layers. Fritts and Dunkerton (1984) and Fritts et al. (2015) explored roles of self-acceleration of GW phase speeds in wave-induced instabilities and momentum deposition. Muraschko et al. (2015) developed a method based on the phase-space WKB theory to accurately compute time-height evolution of wave activity in the time-dependent background flow. Bölöni et al. (2016) showed importance of direct (non-dissipative and weakly nonlinear) interaction between transient GWs and mean flow, extending Muraschko et al. (2015)'s method to the



anelastic airflow system. Kruse and Smith (2018) confirmed predominance of non-dissipative wave-mean interaction due to transience of orographic GWs over the dissipative interaction.

Planetary-scale flows in the middle atmosphere, through which GWs propagate, can exhibit substantial spatial inhomogeneity and transience. Substantially disturbed large-scale flows are often found during sudden stratospheric warming (SSW) events

in association with large planetary wave (PW) activities (e.g., Albers and Birner, 2014; Song and Chun, 2016), and they may result in substantial change in horizontal wavenumbers and frequencies of propagating GWs. This change in GW spectral properties results in spatiotemporal variations of GW pseudomomentum fluxes.

There have been various modeling studies on roles of GWs in SSWs. Richter et al. (2010) showed that source-based GWPs influence PW activities and improves frequency of SSW occurrence in WACCM. Limpasuvan et al. (2012) demonstrated using

WACCM that GW momentum forcing is involved in both SSW initiation and recovery from SSWs. Albers and Birner (2014) discussed roles of GWs in PW resonance before the onset of the 2009 SSW. In recovery phases of SSWs, modeling studies (e.g., Chandran et al., 2013; Limpasuvan et al., 2016) have also reported that combined effects of PWs and GWs are important in generation and evolution of elevated stratopauses (ESs) (Manney et al., 2008). However, given that GW refraction and transient propagation cannot be considered in these models with conventional columnar GWPs, there may be limitations in the

model-based assessment of relative importance between PWs and GWs in evolutions of SSWs and ESs.

Satellite observations have presented evidences of substantial variations of GW activities around SSW onset dates. GW activities are often found to be enhanced in the upper stratosphere before SSW onsets and in high-altitude regions where ESs form in the recovery phase of SSWs (e.g., Yamashita et al., 2013; Thurairajah et al., 2014). These variations of GW activities are also supported by GW-resolving model results for the 2009 SSW (e.g., Yamashita et al., 2010; Limpasuvan et al., 2011).

de Wit et al. (2014) demonstrated substantial change in GW momentum fluxes and forcing in the upper mesosphere around the onset of the 2013 SSW using meteor radar observations over Trondheim, Norway. They showed that the magnitude and evolution of estimated GW momentum forcing are comparable to results from WACCM. However, it is unclear how the two estimates of momentum forcing can be similar, even though the modeled upper-mesospheric winds look quite different from the radar observations. This inconsistency may possibly be attributed to long-distance horizontal propagation of GWs between

the lower atmosphere and the upper mesosphere (e.g., Sato et al., 2009; de Wit et al., 2014; Thurairajah et al., 2017).

The present study explores effects of the 4D ($x$–$z$, $t$) propagation of GWs on distributions of pseudomomentum fluxes, a central quantity in GW-mean flow interaction, for the 2009 SSW. Ray-tracing model for inertia-gravity waves (IGWs) on sphere, whose prototype was used by Song et al. (2017), is employed to compute trajectories and pseudomomentum fluxes of GWs for specified (time-varying) large-scale flows. Diagnosis of mean-flow responses to change in GW pseudomomentum fluxes

is not attempted in this study, since slowly-varying mean-flows are not only modified by GW pseudomomentum but also by the second-order mean pressure fields that can induce mean motions in regions far from localized GW packets (Bühler, 2014). For statistical robustness, ensemble simulations are carried out, similar to experiments for stochastic GWPs (e.g., Dunkerton, 1982; Eckermann, 2011). In each simulation, properties of a monochromatic GW packet at a horizontal grid point are randomly drawn from populations for properties of orographic or nonorographic GWs used in GWPs.





The paper is organized as follows. Section 2 presents formulations of the ray-tracing model. Section 3 describes specification of large-scale flow from the ground to the lower thermosphere. Ensembles for parameterized orographic and nonorographic GWs are presented in Sect. 4. In Sect. 5, ray-tracing simulation results for the 2009 SSW are demonstrated by comparing 4D ($x$–$z$, $t$) and 2D ($z$, $t$) results. Summary and discussion is given in the last section.

## 2 Ray-tracing model for IGWs on sphere

### 2.1 Kinematic wave theory

A wave packet is defined by a group of phase surfaces over distance of the order of a dominant wavelength. A ray is a curve whose tangents coincide with a sequence of wave propagation directions (Landau and Lifshitz, 1975).

Kinematic wave theory (Hayes, 1970) relates the ground-based (observed) frequency $\omega$ and the 3D wavenumber $\boldsymbol{k}$ to a variable $\psi(\boldsymbol{r},t)$ called the phase as follows:

$$\omega(\boldsymbol{r},t) = -\partial\psi(\boldsymbol{r},t)/\partial t, \tag{1}$$

and $\boldsymbol{k}(\boldsymbol{r},t) = \boldsymbol{\nabla_r}\psi(\boldsymbol{r},t),$ (2)

where $\boldsymbol{\nabla_r} = \boldsymbol{e}_\lambda/(r\cos\phi)\partial/\partial\lambda + (\boldsymbol{e}_\phi/r)\partial/\partial\phi + \boldsymbol{e}_r\partial/\partial r$ in the spherical coordinate system; $\boldsymbol{e}_\lambda$, $\boldsymbol{e}_\phi$, and $\boldsymbol{e}_r$ are orthogonal unit vectors in the eastward, northward, and radial directions, respectively; $\lambda$, $\phi$, and $r$ are the longitude, latitude, and radial distance, respectively; $\boldsymbol{r}$ is a position vector; $t$ is time; $\boldsymbol{k}$ can be written as $k\boldsymbol{e}_\lambda + l\boldsymbol{e}_\phi + m\boldsymbol{e}_r$; $k$, $l$, and $m$ are zonal, meridional, and vertical wavenumber components, respectively.

At each $(\boldsymbol{r},t)$, $\omega$ is related to $\boldsymbol{k}$ through a dispersion function ($\Omega$) (Bretherton and Garrett, 1968) given by

$$\omega = \Omega(\boldsymbol{k},\Lambda_1,\cdots,\Lambda_N), \tag{3}$$

where $\Lambda_n$s ($n = 1, \cdots, N$) denote properties of wave propagation medium that vary slowly with respect to phase $\psi(\boldsymbol{r},t)$.

### 2.2 Ray-tracing equations

Time evolutions of position, wavenumber and observed frequency of a wave packet are described as follows:

$$(d\boldsymbol{r}/dt, d\boldsymbol{k}/dt) = (\boldsymbol{\nabla_k}\Omega, -\boldsymbol{\nabla_r}\Omega), \tag{4}$$

and $d\omega/dt = (\partial\Omega/\partial\Lambda_n)\partial\Lambda_n/\partial t.$ (5)

Here, $d/dt$ is the time rate of change following the group velocity ($\boldsymbol{c}_g$) of a wave packet; $\boldsymbol{\nabla_k}$ and $\boldsymbol{\nabla_r}$ are the partial derivatives with respect to wavenumbers and spatial coordinates, respectively; $\boldsymbol{\nabla_k}\Omega = \partial\Omega/\partial\boldsymbol{k} = \partial\Omega/\partial k_i\boldsymbol{e}_i = c_{gi}\boldsymbol{e}_i = \boldsymbol{c}_g$, where $i (= 1,$ 2, or 3) is the summation index that denotes the zonal, meridional, or radial component in order; $\boldsymbol{\nabla_r}\Omega = (\partial\Omega/\partial\Lambda_n)\boldsymbol{\nabla_r}\Lambda_n$.

Equation (4) is isomorphic to the Hamilton's equation for a physical system characterized by a Hamiltonian denoted by $\Omega$. In deriving the $\boldsymbol{k}$ equation [Eq. (4)] from the local time derivative of Eq. (2), a term $c_{gi}(\boldsymbol{\nabla_r}\boldsymbol{e}_i)\cdot\boldsymbol{k}$ shows up, where $\boldsymbol{e}_i$ originates





from $k_i = \boldsymbol{e}_i \cdot \boldsymbol{k} = \boldsymbol{e}_i \cdot (k\boldsymbol{e}_\lambda + l\boldsymbol{e}_\phi + m\boldsymbol{e}_r)$, but it must become zero for the form of ray-tracing equations to be independent of choice of coordinate systems (Ribstein et al., 2015). This constraint gives the $\boldsymbol{k}$ equation shown in Eq. (4).

In the computation of Eqs. (4) and (5), component forms are used (see Appendix A1), and the shallow-atmosphere approximation [$r = a + z \approx a$, where $a$ is the mean radius of the earth, and $z$ ($\ll a$) is the height] is applied (Phillips, 1966; Senf and Achatz, 2011). Under this approximation, $\boldsymbol{\nabla_r} \approx \boldsymbol{\nabla} = \boldsymbol{e}_\lambda/(a\cos\phi)\partial/\partial\lambda + (\boldsymbol{e}_\phi/a)\partial/\partial\phi + \boldsymbol{e}_z\partial/\partial z$, and the magnitude of horizontal wavenumber $|\boldsymbol{k}_h|$ [$= (k^2 + l^2)^{1/2}$] is invariant with respect to the earth's curvature.

## 2.3 Dispersion relation

Dispersion function $\Omega$ is required to compute the ray-tracing equations, and is given by wave dispersion relation.

In the model, the anelastic dispersion relation for IGWs (Marks and Eckermann, 1995) is employed:

$$\hat{\omega}^2 = (\omega - \boldsymbol{k} \cdot \boldsymbol{U})^2 = \frac{N^2(k^2 + l^2) + f^2(m^2 + \alpha^2)}{k^2 + l^2 + m^2 + \alpha^2}, \tag{6}$$

where $\hat{\omega}$ ($> 0$) is the intrinsic frequency; $\boldsymbol{U}$ is the wind vector given by $(U, V, 0)$; $U$ and $V$ are the zonal and meridional wind components, respectively; $N$ is the static stability; $f$ is the Coriolis parameter; $\alpha = 1/(2H)$; $H$ is the large-scale density ($\bar{\rho}$) scale height given by $[-(1/\bar{\rho})\partial\bar{\rho}/\partial z]^{-1}$.

The large-scale flow variables ($U$, $V$, $N^2$, and $\alpha^2$) and $f^2$ correspond to $\Lambda_n$s in Eq. (3), and they are assumed to vary slowly in space and time with respect to GW phases.

## 2.4 Amplitude equation

Time evolution of wave amplitude in slowly varying large-scale flows is described by the wave action conservation law (Bretherton and Garrett, 1968):

$$\partial A/\partial t + \boldsymbol{\nabla} \cdot (\boldsymbol{c}_g A) = -A/\tau_{\mathrm{dis}}, \tag{7}$$

where $A$ is the GW action density [$= E/\hat{\omega}$ ($> 0$)]; $E$ is the phase-averaged GW energy per unit volume; $\tau_{\mathrm{dis}}$ ($> 0$) is the wave dissipation timescale (see Appendix A2 for details).

For computation of wave amplitude along a ray, the conservation law [Eq. (7)] is changed into an equation for vertical action flux $F_A$ ($= c_{gz}A$) after multiplying Eq. (7) by $c_{gz}$:

$$dF_A/dt - F_A/\tau_{\mathrm{def}} = -F_A/\tau_{\mathrm{dis}}. \tag{8}$$

Here, $\tau_{\mathrm{def}}$ is the wave packet deformation timescale (Marks and Eckermann, 1995), at which $|F_A|$ can increase (decrease) for $\tau_{\mathrm{def}} > 0$ ($\tau_{\mathrm{def}} < 0$), and is given by

$$\tau_{\mathrm{def}} = c_{gz}/\left[\partial c_{gz}/\partial t + (c_{g\lambda}\partial c_{gz}/\partial\lambda - c_{gz}\partial c_{g\lambda}/\partial\lambda)/h_\lambda\right.$$
$$\left. + \{c_{g\phi}\cos\phi\,\partial c_{gz}/\partial\phi - c_{gz}\partial(c_{g\phi}\cos\phi)/\partial\phi\}/h_\lambda\right], \tag{9}$$



where $c_{g\lambda}$, $c_{g\phi}$, and $c_{gz}$ are the zonal, meridional, and vertical components of group velocity, respectively; $h_\lambda = a\cos\phi$.

In the wave-mean interaction, the vertical flux of IGW horizontal pseudomomentum ($\boldsymbol{F}_p = c_{gz}\boldsymbol{p}_h = c_{gz}\boldsymbol{k}_h A$, where $\boldsymbol{p}_h$ is the pseudomomentum), rather than the action flux, is a central quantity (Bühler, 2014). Time evolution of $\boldsymbol{F}_p$ along a ray can be obtained by combining results of $\boldsymbol{k}_h$ in Eq. (4) and $F_A$ in Eq. (8). In general, the magnitude and direction of $\boldsymbol{F}_p$ are changed

by refraction due to horizontally varying medium, but $|\boldsymbol{F}_p|$ does not vary owing to curvature terms, as $|\boldsymbol{k}_h|$ is invariant with respect to the curvature (Sect. 2.2).

The action conservation equation [Eq. (7)] is related to conservation of the angular pseudomomentum of IGWs. Combining the component-form equation for $k$ [Eq. (A13)] multiplied by $a\cos\phi$ and the non-dissipative form of Eq. (7) gives the angular pseudomomentum conservation law:

$$d\mathcal{P}/dt + \mathcal{P}\boldsymbol{\nabla}\cdot\boldsymbol{c}_g = \partial\mathcal{P}/\partial t + \boldsymbol{\nabla}\cdot(\boldsymbol{c}_g\mathcal{P}) = 0, \tag{10}$$

where $\mathcal{P}\,(= kAa\cos\phi)$ is the angular pseudomomentum.

## 2.5 Dissipation mechanisms

For dissipation of GWs, two separate processes are employed: Nonlinear wave saturation and molecular diffusion.

Nonlinear saturation is computed by forcing $|\boldsymbol{F}_p|$ not to exceed values for saturated GWs in GW-induced turbulence fields

(Lindzen, 1981). The saturation flux ($F_{p,\text{sat}}$) formulated under the mid-frequency approximation (Senf and Achatz, 2011, SA11 hereafter) is employed and can be written as

$$F_{p,\text{sat}} = \text{Fr}_c^2(\bar{\rho}/2)(|\boldsymbol{k}_h|/N)|U_h - c_p|^3, \tag{11}$$

where $\text{Fr}_c$ is the critical Froude number (McFarlane, 1987, M87 hereafter), and is set equal to $1/\sqrt{2}$ (Hasha et al., 2008); $U_h$ is the horizontal wind parallel to $\boldsymbol{k}_h$ ($U_h = \boldsymbol{U}\cdot\boldsymbol{k}_h/|\boldsymbol{k}_h|$); and $c_p$ is the ground-based phase speed ($c_p = \omega/|\boldsymbol{k}_h|$).

Molecular diffusion is important above the upper mesosphere. Dividing total GW energy equation by $\hat{\omega}$ gives the term $\tau_{\text{dis}}$ due to viscous damping [Eq. (A24)]. In the model, kinematic viscosity is set equal to thermal diffusivity (i.e., $\text{Pr} = 1$, where $\text{Pr}$ is the viscous Prandtl number), and thus complete form of $\tau_{\text{dis}}$ Eq. (A26) is simplified as follows:

$$\tau_{\text{dis}} = 1/\left[2\nu\left(k^2 + l^2 + m^2 + \alpha^2\right)\right], \tag{12}$$

where $\nu$ is the kinematic molecular viscosity.

Kinematic viscosity ($\nu$) is defined as $\mu/\bar{\rho}$, and viscosity $\mu$ is determined as $1.3\times10^{-5}$ kg m$^{-1}$ s$^{-1}$ considering reported values of $\nu$s. Vadas and Fritts (2005) employed $\nu = 6.5$ m$^2$ s$^{-1}$ at $z = 90$ km, and Pitteway and Hines (1963) suggested $\nu = 4$ m$^2$ s$^{-1}$ at the same height. These two values are roughly consistent with the above-mentioned value of $\mu$ for possible range of $\bar{\rho}$ at $z = 90$ km.





### 2.6 Numerical implementation

Time integrations of the ray-tracing equations [Eqs. (4)–(5) or Eqs. (A10)–(A16)] are carried out using the Livermore solver for ordinary differential equation (ODE) with automatic method switching (LSODA) based on the stiffness of an ODE system (Petzold, 1983; Hindmarsh, 1983).

Solutions ($\lambda$, $\phi$, $z$, $k$, $l$, $m$, and $\omega$) of the ray-tracing equations proceed in time over a period of multiples of a timestep ($\delta t$). The timestep ($\delta t$) is determined as 900 s through some tests (see Figs. S1 and S2 in supplement). Large-scale flow variables ($U$, $V$, $N^2$, and $\alpha^2$) are given at an interval of $\Delta t$, and $\Delta t$ should be a multiple of $\delta t$ for proper time marching of the LSODA. In this study, $\Delta t = 1$ h is used.

The solver requires interpolation of $\Lambda_n$s at space and time locations of a GW packet. For spatial interpolation, a local, $C^1$-continuous, tricubic method (Lekien and Marsden, 2005) is employed. The $C^1$ continuity allows for accurate computation of the equation for $\boldsymbol{k}$ that involves the first-order spatial derivatives of $\Lambda_n$s. For temporal interpolation, simple linear interpolation is used, since $\Lambda_n$s are assumed to vary linearly during the time interval ($\Delta t$) of large-scale variables.

Action flux equation [Eq. (8)] is actually an equation for $|F_A|$. Note that $\tau_{\text{def}}$ and $\tau_{\text{dis}}$ required for computation of Eq. (8) do not change the sign of $F_A$.

Dissipation timescale ($\tau_{\text{dis}}$) can be computed along individual rays using ray solutions and large-scale variables at the positions of rays [see Eq. (A26)]. Meanwhile, computation of deformation timescale ($\tau_{\text{def}}$) requires ray-tube information related to spatiotemporal variations of $c_{g\lambda}$, $c_{g\phi}$, and $c_{gz}$ in the neighborhood of a ray [see Eq. (9)]. In the model, $\tau_{\text{def}}$ is estimated through a gridding method: $c_{g\lambda}$, $c_{g\phi}$, and $c_{gz}$ of rays are recorded and accumulated at vertices of grid cells ($\Delta\lambda\Delta\phi\Delta z$) that contain ray paths during $\delta t$. Then, $\tau_{\text{def}}$s are computed at grid points using a finite difference form of Eq. (9) and gridded group-velocity components averaged over overlapped rays. This gridding method is crude compared with the 2D ($z$–$t$) phase-space theory (Muraschko et al., 2015), but it is used to estimate, even roughly, ray-tube effects in the 4D ($\boldsymbol{r}$-$t$) space.

After $\tau_{\text{def}}$ and $\tau_{\text{dis}}$ are obtained, Eq. (8) is computed using the Euler method for the time step $\delta t$. In case that vertical propagation direction is reversed after $\delta t$, the sign of $F_A$ is changed considering the sign of $m$ because $\text{sgn}(F_A) = \text{sgn}(c_{gz}) = -\text{sgn}(m)$. Further details of numerical implementation are described in Appendix A3.

### 3 Large-scale atmospheric flow

Time-varying large-scale flows during the 2009 SSW are specified by combining 6-hourly reanalysis data sets and empirical model results. The reanalysis data are linearly interpolated at an hour interval ($\Delta t = 1$ h). The empirical model results are obtained at the hourly interval using daily 10.7-cm solar flux (F10.7) and 3-hourly geomagnetic activity ($Ap$) indices. Hourly whole-atmospheric flows for ray modeling are obtained by fitting the 3rd-order B-spline curves in the vertical to the time-interpolated hourly reanalysis data and empirical model results in four overlapping layers. Details of the B-spline fit can be found in Song et al. (2018).

The four vertical layers are (i) $p = 10^3$–1 hPa ($z \approx 0$–48 km), (ii) 400–0.1 hPa (7–65 km), (iii) 1–5$\times10^{-4}$ hPa (48–94 km), and (iv) 5$\times10^{-3}$–$10^{-8}$ hPa (84–331 km), respectively. Data used in the four layers are as follows (in order of layer





altitudes): (i) European Centre for Medium Range Forecast (ECMWF) Interim (ERA-Interim, Dee et al., 2011) reanalysis, (ii) Modern-Era Retrospective Analysis for Research and Applications, Version 2 (MERRA-2, Gelaro et al., 2017) reanalysis, (iii) the advanced-level physics high-altitude prototype of the navy operational global atmospheric prediction system (NOGAPS-ALPHA, Eckermann et al., 2009) data, and (iv) empirical model results for geomagnetically quiet-time horizontal winds

(HWM14, Drob et al., 2015) and disturbed horizontal winds (DWM07, Emmert et al., 2008) and temperature (NRLMSISE-00, Picone et al., 2002).

Figure 1 shows latitude-height cross sections of ground-to-space (G2S) zonal wind and temperature at 60°W at 00 UTC on 23 January 2009, one day before the central date (24 January) of the 2009 SSW. The G2S data demonstrate that vertically smooth whole atmospheric wind and temperature profiles can be constructed by fitting the B-spline curves in the vertical

direction to the reanalysis data and empirical model results. In the Northern Hemisphere (NH), the polar-night jet is already reversed above the lower stratosphere north of 60°N, and the weakened eastward jet is tilted from the midlatitudes towards the equator in the lower mesosphere. In association with the jet structure, substantial warming is found in the northern (winter) polar stratosphere, and temperature maximum ($\approx 280$ K) is as high as that in the summer polar stratopause. In the Southern Hemisphere (SH), typical summertime wind and temperature structure is found: Easterly flow in the middle atmosphere below

the upper mesosphere, warm temperature near the polar stratopause region, and coldest temperature and wind reversals in the polar upper mesosphere.

For ray simulations, G2S data at hourly interval are spatially smoothed using the vertical 1-2-1 smoother and horizontal moving averaging on spherical surface. Horizontal averaging is done using variables within the area of a spherical cap centered at every lat-long grid point. A spherical cap is defined by an angle between two lines from the sphere center to the center of

the spherical cap's surface and to the cap's boundary. The angle is set equal to about 2.7°, and for this angle, the area of the spherical cap's surface is equivalent to the area of a circle with a radius of 300 km [$\pi(300 \text{ km})^2$] on a flat surface. Smoothed G2S data are regridded at 2.5°×2.5° horizontal resolution for use in ray simulations.

## 4   GW ensembles

In this study, orographic and nonorographic GWs are separately considered. Properties of orographic GWs are given by the

M87 scheme. Nonorographic GWs are specified based on SA11 and Warner and McIntyre (1996) (WM96 hereafter).

### 4.1   Orographic GWs

Orographic GWs (OGWs) are assumed to be excited when low-level horizontal winds are strong, and vertical parcel displacements due to subgrid topography are large (M87).

The vertical displacement ($h_m$) is given by $2\sigma_h$, where $\sigma_h$ is the standard deviation of the subgrid-scale topography. Reynolds

stress due to stationary OGWs ($\omega = 0$ and $c_p = 0$) is given by $(k_{ho}/2) \min(h_m^2, U_s^2/N_s^2) \rho_s N_s U_s$, where $k_{ho}$ is the horizontal wavenumber [$= 2\pi/(100 \text{ km})$], and $\rho_s$, $N_s$, and $U_s$ are air density, stability, and horizontal wind magnitude, respectively,





averaged within the source layer between the ground to the $h_m$. Grid- and subgrid-scale topography are obtained by averaging and regridding NCAR Community Earth System Model (CESM) auxiliary data with horizontal resolutions close to $2.5° \times 2.5°$.

Directions of horizontal wavenumber vectors are set opposite to horizontal wind vectors averaged within source layer. OGWs are launched at the top of the source layers.

## 5  4.2  Nonorographic GWs

For nonorographic GWs (NGWs), three 14 discrete wave schemes as in SA11 are considered. One is a modified version of SA11, and the other two are derived from the empirical spectra of WM96.

The empirical GW energy spectrum ($\hat{E}$) in WM96 is given by a separable function of $m$, $\hat{\omega}$, and the azimuth angle ($\varphi$) [i.e., $\hat{E} = E_0 A(m) B(\hat{\omega}) \Phi(\varphi)$]. The pseudomomentum flux spectrum $\bar{\rho} c_{gz} \hat{E}(m, \hat{\omega}, \varphi) \boldsymbol{k} / \hat{\omega}$ can be written as a function of $k$, $\omega$, and

$\varphi$ by multiplying the spectrum by the Jacobian factor ($J = m/|\boldsymbol{k}|$). This spectrum is discretized to obtain 14-wave schemes through numerical integrations around appropriately chosen $k$s and $\omega$s for 14 sets of $\varphi$ and $c_p$ as in SA11 in a quiescent atmosphere with a specified stability (0.01 rad s$^{-1}$). Two 14-wave WM96 schemes are obtained by using two different values (1 and 5/3) of $p$ in the spectrum $B(\hat{\omega})$ given by $B_0(p)\hat{\omega}^{-p}$. NGWs are launched at every horizontal grid point at $z = 6.8$ km near 400 hPa.

Figure 2 illustrates angular histograms of spectral properties and Reynolds stress in the three 14-wave NGW schemes. In these schemes, horizontal propagation directions ($\varphi$s) and ground-based phase speeds ($c_p$s) are given for each of 14 GWs, and they are identical to those in SA11. The horizontal wavelength ($\lambda_h$) in SA11 ranges from 385 to 596 km. In WM96 with $p = 5/3$ (WM96a), the range of $\lambda_h$ is broader (309–782 km) compared with SA11, and in WM96 with $p = 1$ (WM96b), the range is much broader (128–942 km).

Each GW has identical amount of Reynolds stress in the three schemes. For this, the stresses for GWs with $c_p > 20$ m s$^{-1}$ are reduced in SA11, and integration intervals of $k$ and $\omega$ for the spectra in WM96 are appropriately adjusted. As a result, NGWs with $c_p$s smaller (larger) than 20 m s$^{-1}$ have Reynolds stress of the order of $10^{-3}$ ($10^{-5}$) N m$^{-2}$. Reynolds stresses exhibit slightly more westward flux, but they are almost isotropic. The total magnitudes of Reynolds stresses used in this study are $3.6 \times 10^{-3}$ N m$^{-2}$ in the eastward, northward and southward directions and $4.1 \times 10^{-3}$ N m$^{-2}$ in the westward direction.

These magnitudes are comparable to the total momentum flux of $3.75 \times 10^{-3}$ N m$^{-2}$ in each Cardinal direction at 450 hPa that is employed in the ECMWF model (Orr et al., 2010). Details of GW properties shown in Fig. 2 are found in Table S1 in the supplement.

## 4.3  Generation of GW ensembles

In ray simulations, single GW packet, stochastically chosen from GW source ensembles, is launched at a horizontal grid point

where GWs are supposed to be generated. This approach is computationally efficient, allowing for statistical significance tests for differences between ray simulations.

The OGW scheme (M87) launches single OGW at a horizontal grid point, but NGW schemes usually specify multiple GWs. Hence, GW ensembles are separately generated for OGWs and NGWs. For OGWs, the vertical displacement $h_m$ is assumed



to be given by $s_f\sigma_h$, where the scale factor $s_f$ has a uniform probability distribution between 1 and 3 around its default value 2. For NGWs, single GW is selected with uniform chance from 14 discrete waves. GW ensembles are precomputed using a random-number generator for reproducibility.

Figure 3 demonstrates horizontal distributions of stochastic parameters, zonal OGW and NGW pseudomomentum fluxes

($F_p$s) at individual launch levels for a particular ensemble member, and the ensemble averages of the zonal $F_p$s at 00 UTC on 20 January 2009, four days before the 2009 SSW onset. The stochastic parameters for OGWs and NGWs are the displacement scale factors ($s_f$) and wave IDs (1–14), respectively. For each of GW source schemes, 20 ensemble members are generated.

For OGWs, $s_f$ varies randomly between 1 and 3 at grid points where $\sigma_h$ is nonzero. In the mid-latitude tropospheric eastward jet regions, westward OGW $F_p$ are large and may reach about $-1$ N m$^{-2}$ over major mountainous regions: The Alps,

the Tibetan plateau, the Rockies, and the Andes. Large eastward OGW $F_p$s are found in the higher-latitude regions such as Greenland and the Antarctica. Zonal $F_p$s in each OGW ensemble member have locally substantial deviations from the ensemble mean (Fig. 3c) in the major mountain areas. Maximum value of the standard deviation from the ensemble mean is about 0.7 N m$^{-2}$ and found in the Tibetan areas.

For NGWs, wave IDs of 1–14 are randomly spread on the globe. The magnitude of ensemble-averaged zonal NGW $F_p$s is

$O(10^{-3}$ N m$^{-2})$. The horizontal distribution of zonal $F_p$s in each NGW ensemble member (Fig. 3e) looks noisy, but characteristic structure is more clearly revealed in its ensemble average (Fig. 3f). The sign of zonal NGW $F_p$s is generally opposite to that of the tropospheric zonal flows.

## 5 Results

GW ray simulations are carried out for the time period of 25 days from 00 UTC on 8 January 2009 to 00 UTC on 2 February

2009 for the 20 OGW and NGW ensemble members.

For each GW ensemble member, two kinds of simulations are carried out: Four-dimensional (4D) ($x$–$z$, $t$) and two-dimensional (2D) ($z$, $t$) experiments. In 4D experiments, GW rays propagate horizontally as well as vertically in spatially varying background media, but in 2D, rays are allowed to propagate only in the vertical direction. In both 4D and 2D, GW rays propagate through time-varying flows, and therefore modulations of the observed frequencies of GWs occur. In 4D cases for M87 and

SA11, additional simulations where $\tau_{\mathrm{def}} = 0$ in the amplitude equation are carried out to see ray-tube effects. In all simulations, 3-h averaged gridded outputs are generated. GW rays are launched every 3 h, and 3-day old rays are eliminated. These launch interval and ray life time are chosen considering computational time, the time scale of the large-scale flow, and elapsed time for rays launched in the troposphere to reach the upper mesosphere and lower thermosphere (UMLT) (see Figs. S1–S3).

### 5.1 Zonally-averaged GW properties

Figure 4 shows latitude-height distributions of zonal-mean zonal wind and ensemble averages of zonal-mean zonal $F_p$s of OGWs and three NGWs in the 4D and 2D experiments at 00 UTC on 20 January 2009. The NH polar vortex is not reversed yet but being weakened. Transparently shaded areas indicate regions where differences between 4D and 2D are not statistically





significant. The signs of the zonal-mean zonal $F_p$s below $z = 80$ km are overall similar in the 4D and 2D and seem to be related to structure of the zonal wind, but they become different from each other above $z = 80$ km in the NH polar regions.

Statistically significant differences between the 4D and 2D are found in five regions. (i) In the latitude-height region of $40°N–70°N$ and 40–50 km, westward $F_p$s in the 4D are about 10 (28) times enhanced when compared with 2D results for OGW, $NGW_{SA11}$, and $NGW_{WM96a}$ ($NGW_{WM96b}$). (ii) In the NH polar UMLT ($70°N–80°N$ and 85–100 km), the magnitude of eastward $F_p$s in the 4D is 1.5–5 times larger than that of westward $F_p$s in the 2D for both OGWs and NGWs. This result implies that 4D results may better explain the mesospheric cooling around the SSW central dates, compared to the 2D, given that the cooling can be induced by eastward GW momentum forcing in the UMLT. (iii) In the upper troposphere and lower stratosphere (UTLS) above the tropospheric jets, westward $F_p$s of NGWs are 1.6–2.6 time larger in the 4D. (iv) In the SH mesosphere, eastward $F_p$s of NGWs in the mid-latitudes are reduced by more than half in the 4D. (v) Eastward $F_p$s of OGWs at $z = 30–80$ km around $70°S$ are 5–6 times enhanced in the 4D. As a result, the magnitude of zonal $F_p$s in the middle atmosphere and their structure in the UMLT can be substantially changed in the 4D where horizontal propagation and refraction are allowed, even though $\boldsymbol{F}_p$s are given identically at source levels in the 4D and 2D experiments.

Additional 4D OGW and $NGW_{SA11}$ experiments (Fig. S4) where no ray-tube effects are considered ($\tau_{def} = 0$) give similar results to those with $\tau_{def} \neq 0$ shown in Fig. 4, except for the NH upper stratosphere and lower mesosphere (USLM). Statistically significant differences between nonzero and zero $\tau_{def}$s are found in relatively narrow regions around the NH jet core ($75°N$ and 40 km at 00 UTC on 20 January 2009) (see Fig. S4). It is interesting that ray-tube effects become important in regions near the jet where the large-scale winds vary rapidly in space (see Sect. 5.2). In these regions, the magnitude of westward $F_p$s when $\tau_{def} \neq 0$ is reduced by less than half compared with when $\tau_{def} = 0$. These differences are localized compared with the differences between the 4D and 2D experiments, which may indicate that ray-tube effects be relatively limited. However, it should be noted that ray simulations in this study may underestimate ray-tube effects, given that horizontal spread of GW fields that emanate from local sources cannot be considered in the current ray simulations where single GW ray is launched at a grid point.

Figure 5 shows latitude-height distributions of zonal-mean zonal wind and ensemble means of zonally-averaged zonal wavenumbers ($k$s) for OGWs and NGWs in the 4D and 2D experiments at 00 UTC on 20 January 2009. It is clear that the sign of zonal-mean $k$s is overall the same as that of the zonal $F_p$s shown in Fig. 4, except for some regions in the NH USLM. This similarity in the signs of $k$s and zonal $F_p$s implies that the zonal-mean zonal $F_p$ (Fig. 4) is mostly due to upward propagating GWs, since the sign of zonal $F_p$ ($= kF_A$) is determined by the sign of $k$ alone in case that $c_{gz} > 0$ and thus $F_A > 0$. In the NH USLM, however, GWs in the 4D are found to propagate downward in some areas, and positive $k$s at $z = 30–50$ km are related to the GWs with negative $c_{gz}$ and westward $F_p$ (see Sect. 5.2 for details).

Statistically significant differences of zonal-mean $k$s between the 4D and 2D are also found in similar regions to those of zonal-mean zonal $F_p$s shown in Fig. 4. In the NH polar UMLT, the sign of $k$s is reversed between the 4D and 2D, and the magnitude of positive $k$s in the 4D is 1.2–6.3 times larger than that of negative $k$s in the 2D for both OGWs and NGWs. In the UTLS, negative $k$s of NGWs are 1.4–2.4 times larger in the 4D. In the SH mesosphere, positive $k$s of NGWs in the mid-latitude regions are reduced by about half in the 4D, and positive $k$s of OGWs around $70°S$ are about 1–3 times enhanced in the 4D.



These changes in $k$s in the 4D with respect to the 2D are roughly similar to those in the zonal $F_p$s shown in Fig. 4. This result indicates that differences in the zonal $F_p$ between the 4D and 2D experiments can be accounted for to a substantial degree by changes in the $k$s between the 4D and 2D.

Time rate of change of $k$s along rays is determined by the five forcing terms [Eqs. (A13) and (A17)]. Among the 5 terms, the two zonal shear forcing terms [$-k/(a\cos\phi)\partial U/\partial\lambda$ and $-l/(a\cos\phi)\partial V/\partial\lambda$] and curvature term ($lc_{g\lambda}\tan\phi/a$) are predominant in most cases. Thermodynamic forcing terms are usually 2–3 order of magnitude smaller (see Fig. S5).

Figure 6 shows latitude-height distributions of the total and three major forcing terms in the $k$ equation for OGWs and NGW$_{SA11}$s in the 4D experiment at 00 UTC on 20 January 2009. Note that the forcing terms in the $k$ equation in the 2D are all zero. For NGWs, results for SA11 are presented since the other NGW schemes give roughly similar results. It is clear that the magnitude of the curvature term is as large as the two zonal shear forcing terms in the mid-latitudes as well as the polar regions, which supports importance of the curvature term presented by Hasha et al. (2008). In the UTLS, the total forcing term for NGWs is generally negative above the tropospheric jet cores (60°S–40°S and 20°N–50°N) where the negative $k$s are predominant and enhanced in the 4D, and the negative total forcing is due mainly to the zonal shear term of zonal winds and curvature effects. For OGWs around 70°S, the enhancement of the positive $k$ in the stratosphere is attributed to curvature effects. For NGWs, curvature effects are predominant over the other two major forcing terms in the SH stratosphere where winds are steady, but it becomes a little smaller than the two zonal shear forcing terms in the NH where the polar vortex varies rapidly in space and time.

Structure of the three major forcing terms in the $k$ equation (Fig. 6) is different from that of $k$s (Fig. 5) except for the NH USLM. This difference may occur in relation to space and time propagation of GWs, since certain $k$s substantially changed in some regions may not be changed a lot as GWs propagate to the other regions. In fact, positive $k$s of NGWs with eastward $F_p$ in the UTLS of the SH mid- to high-latitude regions are increased by positive zonal shear terms for the zonal wind and curvature terms (Fig. S5). The increase in positive $k$s enhances eastward $F_p$ in the SH UTLS. However, the eastward $F_p$s of NGWs is reduced in the SH middle atmosphere, even though total forcing in the $k$ equation for NGWs with eastward $F_p$ is positive (Fig. S5). It is seen that the GWs with enhanced eastward $F_p$ would be dissipated through the saturation as they propagate northward toward the SH mid-latitude USLM. This dissipation results in reduction in eastward $F_p$ and positive $k$ in the SH mesosphere (Figs. 4–5). In contrast, distributions of $k$s and major forcing terms (Fig. 6) are correlated with each other in the NH USLM, which indicate that structure of $k$s in the NH USLM can locally be generated by forcing terms around $z = 40$ km.

Figure 7 shows latitude-height distributions of zonal-mean zonal wind and ensemble means of zonally-averaged meridional wavenumbers ($l$s) for OGWs and NGW$_{SA11}$s in the 4D and 2D experiments at 00 UTC on 15 and 20 January 2009. It is found that structural difference of $l$s between the 4D and 2D is substantial for both OGWs and NGWs, and is more significant than the difference in structure of $k$s. This result may be related to larger meridional variations of the large-scale flow than its zonal variations, given that the time rates of change of $l$ and $k$ are determined by meridional and zonal variations of the large-scale flow, respectively.

In the SH, large positive $l$s of OGWs appear around 70°S in the 4D compared with the 2D on both the two dates, and these are due to the positive meridional shear forcing term related to zonal wind [$-(k/a)\partial U/\partial\phi$], where $k > 0$, and $\partial U/\partial\phi < 0$



(see Fig. S6). In the NH USLM, positive (negative) $l$s of OGWs in the 4D appears roughly south (north) of the eastward jet axis on 15 January, but significantly enhanced positive $l$s are predominant on 20 January. The signs of $l$s across the jet on 15 January means northward (southward) propagation of OGWs relative to large-scale meridional winds. Therefore, OGWs in the 4D can in fact be converged along the jet axis in the NH USLM, as long as the meridional variations of meridional winds are

not significant. On 20 January, as the jet is moved towards $80°$N, it is seen in the 4D that positive $l$s of OGWs and poleward propagation of OGWs become predominant south of the jet axis. In the 2D, structure of $l$s of OGWs in the NH does not seem to exhibit significant responses to spatiotemporal variations of the large-scale flow.

Structure of $l$s for NGWs is coherent with the large-scale flow structure in the 4D, but it seems more or less random in the 2D. In the SH, $l$s of NGWs in the 4D are overall positive in the high-latitude stratosphere above $z = 10$ km and in the

mid-latitude middle atmosphere above $z = 30$ km, and $c_{g\lambda}$ is overall negative due to the westward winds. This explains why the curvature term $(lc_{g\lambda} \tan\phi/a)$ in the $k$ equation is positive for NGWs in the SH ($\phi < 0$) middle atmosphere, as shown in Fig. 6h. The positive $l$s of NGWs in the SH are due mainly to the forcing term related to the meridional shear of zonal winds and curvature term $(-kc_{g\lambda} \tan\phi/a)$ in the $l$ equation (see Fig. S6). Positive $k$s above $z = 40$ km (Fig. 5) and change in the sign of $\partial U/\partial\phi$ around the axis of the westward winds ($\partial U/\partial\phi < 0$ south of the axis and $\partial U/\partial\phi > 0$ north of the axis) make

the meridional shear forcing term in the $l$ equation positive in the SH polar USLM and negative in the SH mid-latitude USLM. In the NH USLM, as in OGWs, structure of the $l$s on 15 January may indicate latitudinal convergence of NGW rays along the axis of the stratospheric jet. On 20 January, positive $l$s are predominant in the NH USLM, and NGWs south of $80°$N propagate northward.

Figure 8 shows latitude-height distributions of zonal-mean zonal wind and ensemble means of zonally-averaged number of

GW packets for OGWs and NGW$_{SA11}$s in the 4D and 2D experiments at 00 UTC on 15 and 20 January 2009. The number of GW packets corresponds to the ray counter used for averaging in the gridding method. Comparison between the 4D and 2D indicates that there is convergence of GW packets in the 4D along the jet axis in the NH USLM and in the NH polar stratosphere on 15 January for both OGWs and NGWs. In the latitude-height region of $40°$N–$70°$N and 40–90 km on 15 January, zonally averaged number of rays for OGWs (NGWs) is about 0.9 (1.9) in the 4D, and it is 1.7 (1.3) times larger than in the 2D. In

this region, GWs with westward $F_p$ outnumber GWs with eastward $F_p$, and therefore the ray convergence may increase the westward $F_p$. In the NH mid- to high-latitude lower thermosphere ($30°$N–$90°$N and 120–140 km), zonally averaged number of rays for OGWs (NGWs) is about 2.5 (1.9) times larger in the 4D on 15 January. This difference is due mainly to OGWs and NGWs with eastward $F_p$ (not shown).

As the eastward jet in the stratosphere moves towards the North pole on 20 January, the number of GW packets in the 4D

increases in the NH polar mid- to upper-stratosphere ($50°$N–$80°$N and 30–50 km). In this region, zonally averaged number of rays for OGWs (NGWs) is about 0.9 (2.3) in the 4D, and it is 1.7 (1.1) times larger than in the 2D. As in 15 January, GWs with westward $F_p$ are predominant in the NH stratosphere, resulting in some enhancement of westward $F_p$s. In the NH lower thermosphere ($30°$N–$90°$N and 120–140 km), zonally averaged number of rays for both OGWs and NGWs is about 1.8 times larger in the 4D. This increase is mostly attributed to OGWs and NGWs with eastward $F_p$s, as in 15 January.





GWs generally propagate more to the thermosphere in the 4D. Even though the eastward winds are still large in the NH middle atmosphere on 15 and 20 January, both OGWs and NGWs with eastward $F_p$ (i.e., $k > 0$ and $c_{p\lambda} - U > 0$, where $c_{p\lambda}$ is the ground-based zonal phase speed) can propagate better to the thermosphere in the 4D, being less dissipated in the middle atmosphere. There seems a tendency in the 4D for GWs to better elude critical-level filtering or nonlinear saturation in

the lower atmosphere. This tendency may be attributed to larger degree of freedom in propagation, associated with change in wavenumber directions due to refraction and curvature effects that can occur before either filtering or saturation is initiated as GWs approach critical levels.

Similar results can also be found for NGWs in the SH. More NGW packets in the 4D are found in the SH middle atmosphere, and they are related to reduced restriction in the propagation of GWs with eastward $F_p$ towards the middle atmosphere. Also,

as GWs propagate better toward the upper atmosphere, GW packets (with eastward $F_p$) in the 4D look vertically more spread near $z = 90$ km in the SH where the zonal-mean zonal wind is reversed. This spread in the 4D compared with the 2D implies that GWs in the 4D may better avoid filtering without being trapped in a narrow vertical layer close to critical levels.

Convergence (or focusing) of GW packets may have some effects on distribution of the GW pseudomomentum fluxes. Song and Chun (2008) emphasized this effect as an important mechanism that account for differences between ray-based and

columnar GWPs. Discussion regarding Fig. 8 indicates that convergence or divergence (spread of GW packets) effects in the 4D would be smaller than a factor of 2 in terms of the magnitude of $F_p$s. These convergence and divergence effects are, however, likely relatively small compared with impacts due to change in horizontal wavenumbers shown in Figs. 4–5, although the refraction effects would not entirely lead to local change in mean flows (see Sect. 5.2).

### 5.2 Horizontal distributions of GW characteristics

Figure 9 shows longitude-latitude distributions of zonal and meridional winds and ensemble averages of zonal pseudomomentum fluxes ($F_p$s) and zonal and vertical wavenumbers ($k$s and $m$s) for OGWs at $z = 38$ km in the 4D and 2D experiments at 00 UTC on 20 January 2009. Zonal and meridional winds demonstrate that planetary-scale waves of zonal wavenumber 2 are significantly enhanced in the NH stratosphere. The planetary waves are accompanied by large spatial gradients of horizontal winds in the NH mid- to high-latitude regions. As shown in the previous section, zonal $F_p$s and $k$s can be significantly changed

in association with zonal gradients of horizontal winds.

Horizontal structure of zonal $F_p$s is significantly different between the 4D and 2D. First of all, zonal $F_p$s of OGWs in the 4D are widespread in the NH mid- to high-latitude regions, and they are significantly enhanced in some particular areas where large zonal gradients of horizontal winds appear. In the 4D, nonzero zonal $F_p$s are newly found over the Western Europe, north of Northern Europe, north of the Kamchatka peninsula, west of North America, and over Greenland. This regional difference

in nonzero zonal $F_p$s indicates effects of horizontal propagation. In polar regions and west of North America, zonal $F_p$s of OGWs become eastward in the 4D. These eastward $F_p$s are related to positive $k$s induced by refraction or curvature effects. In the 2D, eastward $F_p$s of OGWs appear only over Greenland and some regions along the Antarctic coastlines where $F_p$s are eastward from launch levels.



Increase in the magnitudes of zonal $F_p$s and $k$s appears together in narrow areas between southward and northward winds (80°E and 80°W–140°W around 30°N–70°N), between westward and eastward winds (20°E–60°E and 30°N), and along the polar eastward jets. This enhancement is spatially correlated with the two zonal shear forcing and curvature terms in the $k$ equation (not shown), and therefore it is thought of as being locally induced by the wind shear and curvature terms. In these

narrow areas, the magnitudes of $k$s and $m$s become $O(10^{-4}–10^{-3}$ rad m$^{-1})$ and $O(10^{-2}–10^{-1}$ rad m$^{-1})$, respectively. That is, zonal and vertical wavelengths can be as small as $O(1–10$ km$)$ and $O(10–100$ m$)$, respectively. In some areas between southward and northward winds (60°E and 130°W), $m$s are positive (i.e., $c_{gz} < 0$), positive $k$s are large, and as a result, westward $F_p$s are enhanced. This situation shows that large increase in $F_p$s can occur locally, as GWs propagate downward experiencing substantial horizontal refraction.

Horizontal distributions of zonal $F_p$s, and zonal and vertical wavenumbers ($k$s and $m$s) for NGW$_{SA11}$s at $z = 38$ km at 00 UTC on 20 January 2009 (Fig. 10) can also be accounted for in similar ways as in case of OGWs shown in Fig. 9. Statistically significant difference in zonal $F_p$s between the 4D and 2D are mostly found in the NH mid- to high-latitude regions. Similar to OGWs, zonal $F_p$s of NGWs are significantly enhanced in regions where the zonal gradients of horizontal wind components and curvature effects are large (not shown). Enhancement of westward $F_p$s near 60°E (north of Canada) is largely due to the

zonal shear term of zonal (meridional) winds. Along narrow regions where large zonal gradients of meridional winds appear (Fig. 10b), positive $k$s are substantially increased, $m$s are positive (i.e., $c_{gz} < 0$), and as a result, zonal $F_p$s become westward. In these regions, westward $F_p$s become significantly large, and zonal and vertical wavelengths of NGWs can also be $O(1–10$ km$)$ and $O(10–100$ m$)$, respectively, owing to horizontal refraction related to the zonally varying meridional wind.

Figure 11 shows longitude-latitude distributions of zonal and meridional winds and ensemble averages of horizontal com-

ponents of the group velocity ($c_{g\lambda}$ and $c_{g\phi}$), horizontal components of the intrinsic group velocity ($c_{g\lambda} - U$ and $c_{g\phi} - V$), vertical component of the group velocity ($c_{gz}$), and ratio of intrinsic frequency to Coriolis parameter ($\hat{\omega}/|f|$) for OGWs at $z = 38$ km in the 4D experiment at 00 UTC on 20 January 2009. Distributions of horizontal components of the group-velocity and horizontal winds look similar to each other, but in fact they exhibit distinct differences, as shown in the intrinsic group velocity. Over the Tibetan plateau and North America in which the eastward jet cores are located, both components of the horizontal

intrinsic group velocity are not close to zero. Therefore, in these regions, OGWs propagate relative to the horizontal winds.

Meanwhile, in narrow regions around 60°E and 130°W, the meridional wind component varies rapidly, and both components of the horizontal intrinsic group velocity are roughly close to zero, when judged from zero contour lines for both intrinsic group velocity components in the narrow regions. Also, in the elongated areas along the axis of the polar eastward jet (Arctic areas centered on the longitudes of 0° and 180°), there are large zonal gradients of the zonal wind, and horizontal intrinsic group

velocity is nearly zero. Therefore, OGW packets roughly move at the large-scale horizontal wind velocity in these narrow and elongated areas. That is, OGW packets behave like tracers advected by the horizontal winds. Vertical group velocity components are at least one order of magnitude smaller than horizontal components of the intrinsic group velocity [$O(1–10$ m s$^{-1})$] in these narrow and elongated regions. Near 60°E, $c_{gz}$ is close to zero. Positive $c_{gz}$s of $O(1$ m s$^{-1})$ are found at a few grid points near 130°W, but $c_{gz}$s (with negative and positive signs) are largely $O(0.1$ m s$^{-1})$. Along the polar eastward jets,

$c_{gz}$s are also $O(0.1$ m s$^{-1})$.





Equation (13) shows the magnitude of the 3D intrinsic group velocity as a function of wavenumbers and intrinsic frequency under the Boussinesq approximation ($m^2 \gg \alpha^2$):

$$|\hat{\boldsymbol{c}}_g| = \frac{1}{\kappa}\sqrt{\frac{(N^2 - \hat{\omega}^2)(\hat{\omega}^2 - f^2)}{\hat{\omega}^2}}, \tag{13}$$

where $\kappa$ is the magnitude of the 3D wavenumber vector ($\kappa = \sqrt{k^2 + l^2 + m^2}$). From Eq. (13), it is clear that the magnitude of

the intrinsic group velocity approaches zero in case $\kappa$ significantly increases.

It is already seen that the magnitude of zonal and vertical wavenumbers (Fig. 9) are substantially increased in the narrow and elongated regions around 60°E and 130°W and along the axis of the polar eastward jets. In these regions, $(c_{g\lambda}, c_{g\phi}) \approx (U, V)$, as shown in Fig. 11. For small $|\hat{\boldsymbol{c}}_g|$, intrinsic frequency need not to be necessarily small [Eq. (13)], although the ratio $\hat{\omega}/|f|$ becomes close to the limiting value ($\sqrt{2}$) as the magnitude of wavenumber approaches infinity (Bühler, 2014). The ratios in

fact have broad range of values in the narrow regions. The ratios are quite small (1–2) over western Greenland, but they are quite large (10–20) north of Alaska and north of Northern Europe.

The significant enhancement of zonal $F_p$s ($= k c_{gz} A$) in the narrow and elongated areas implies increase in zonal GW pseudomomentum ($kA$), since $c_{gz}$s are not particularly increased. Increase of the zonal pseudomomentum in these regions where the intrinsic group velocity is quite small indicates that there is possibility of occurrence of the wave capture (Bühler

and McIntyre, 2005) in the narrow and elongated areas during the evolution of the SSW event. When $|\hat{\boldsymbol{c}}_g|$ is small in the highly strained large-scale flow, GW packets behave like tracers, and their shape is also substantially stretched, which leads to increase of wavenumbers and thus pseudomomentum. For GW packets in slowly varying mean flows, the change in the pseudomomentum should be balanced by change in a quantity called impulse defined by mean-flow vortex structure away from GW packets, unless there are external forces (Bühler, 2014). Hence, the enhanced pseudomomentum of GWs captured

by distorted mean flows can cause change in vortical motions far from GW packets. The surged pseudomomentum in the narrow areas may not lead to significant local change in mean flows even when the GWs are dissipated, as described by Bühler (2014).

As shown in Figs. 4–5, zonally-averaged eastward $F_p$s in the UMLT are enhanced in the 4D, and they are related to positive $k$s. At $z = 92$ km, differences in zonal $F_p$s of NGWs between the 4D and 2D appear over broad longitudes in the NH polar

region north of 60°N (Fig. S7). In the 4D, eastward $F_p$s in the NH polar region are correlated with positive $k$s, and $m$s are negative ($c_{gz} > 0$). The positive $k$s are not locally induced by zonal shear or curvature terms near $z = 92$ km (not shown), and they seem to be gradually acquired, as NGWs propagate upward through the middle atmosphere.

## 5.3 Time variations of pseudomomentum fluxes

Figure 12 shows time-height cross-sections of zonal-mean zonal wind and ensemble averages of zonal $F_p$s of OGWs and

NGW$_{SA11}$s averaged between 30°N and 90°N in the 4D and 2D experiments from 8 January to 2 February 2009. Time-height distributions of zonal $F_p$s are also quite different between the 4D and 2D. First, westward $F_p$s in the 4D are significantly enhanced in the upper stratosphere from 13 January, about ten days before the onset date (24 January). Increase of westward $F_p$s is stronger in OGWs than in NGWs. Second, there are larger eastward $F_p$s of OGWs and NGWs in the lower thermosphere





in the 4D from the early stage of the SSW evolution. Third, eastward $F_p$s are substantially enhanced in the upper mesosphere a few days earlier than the onset date in the 4D. Fourth, eastward $F_p$s of OGWs are increased in the 4D around $z = 40$ km in the recovery phase after the onset.

Enhanced westward $F_p$s in the upper stratosphere before onset are clearly related to the surge of pseudomomentum associated with GWs captured by vortical mean flows. The other enhancements in the magnitude of zonal $F_p$s in the 4D, however, are not related to the wave capture, and they are likely due to GWs in the 4D that propagate upward better avoiding filtering or saturation in the lower atmosphere. In the 2D, significant eastward $F_p$s in the lower thermosphere are not found before the onset, which may indicate that vertical propagation is quite restrictive in the 2D. Eastward $F_p$s of NGWs in the USLM near the onset occur in both the 4D and 2D, but eastward $F_p$s of OGWs appear only in the 4D. The eastward $F_p$s are increased around $z = 40$ km below the recovered eastward jets a few days after the onset date in the 4D for both OGWs and NGWs. For OGWs, these enhanced eastward $F_p$s are induced by upward propagating OGWs with eastward $F_p$s from source layers, since westward winds prevails from the ground to $z = 40$–50 km for several days after the onset. These enhanced eastward $F_p$s, if they exist, may induce more rapid recovery of the stratospheric jets, accelerating downward movement of the ES.

## 6   Summary and discussion

Effects of realistic propagation of parameterized GWs on GW pseudomomentum fluxes are investigated using a global ray-tracing model for the 2009 SSW event. Two kinds [4D ($x$–$z$, t) and 2D ($z$, t)] of ray simulations are carried out to understand propagation effects for 20 ensemble members of OGWs and NGWs for the time period of 25 days from 8 January to 2 February 2009. In each ensemble member of OGWs and NGWs, single GW packet is launched at a horizontal grid point, and properties (wavelength, phase speed, propagation direction, and Reynolds stress) of GW packets are randomly chosen from a precomputed set of parameters made based on previous GWP studies for OGWs and NGWs.

Global ray-tracing model used in this study is composed of two parts: Ray-tracing and amplitude equations. Ray-tracing equations are formulated considering the curvature effects on spherical earth, and they compute trajectory of GW packets and refraction due to spatiotemporal variations of the large-scale flow. Time evolution of vertical flux of GW action flux is computed using the amplitude equation. In the amplitude equation, ray-tube effects associated with geometry of neighboring rays are considered by evaluating group velocity components of GW packets at grid points. For dissipative processes, nonlinear saturation and molecular viscosity are computed along ray trajectories. These dissipations only act on the action flux without affecting GW propagation.

In realistic 4D propagation, horizontal refractions related to large-scale wind shear and curvature effects are essential compared with spatial gradients of thermodynamic large-scale properties such as stability and density. Latitude-height structure of the zonal pseudomomentum fluxes ($F_p$s) is overall similar to that of the zonal wavenumbers except for the NH UMLT region. This structural agreement indicates that most of GWs propagate upward in the ray simulations in this study. The magnitude of zonal $F_p$s, however, is locally quite different between the 4D and 2D experiments. In the 4D, westward $F_p$s are enhanced in the UTLS in both hemispheres and in the NH upper stratosphere, and eastward $F_p$s are reduced in the SH USLM. In the NH





UMLT, the sign of zonal $F_p$s are reversed between the 4D and 2D. It is seen that eastward $F_p$s in the NH UMLT for the 4D are due to GWs that can propagate upward better avoiding critical-level filtering and saturation in the lower atmosphere. As GW packets are refracted, GW packets can be converged around the axis of the stratospheric jet. Locally increased number of GW packets can have some effects on the zonal $F_p$s in a factor of 2 or less in terms of magnitude. Ray-tube effects are present in the NH upper stratosphere where planetary-scale wave activities are large, but they are not significant in the other regions.

In the NH upper stratosphere, westward $F_p$s are significantly enhanced in the 4D experiment along narrow and elongated areas in the mid- to high-latitude regions where spatial variations of the large-scale winds are substantial in association with large planetary-wave activities. The significant enhancement in zonal $F_p$s is due mainly to the horizontal refraction related to the horizontal wind shear and curvature effects. In the narrow regions, intrinsic group velocity components are quite small, which means that GWs travel roughly following the large-scale winds. This result indicates that the wave capture phenomena may occur along the meandering eastward jets during the evolution of the SSW. For GW packets in slowly-varying mean flows, change in GW pseudomomentum due to refraction and curvature terms is balanced by change in the impulse defined by the structure of nearby mean-flow vortices. The enhanced $F_p$s of captured GWs may not affect directly local mean flows where GW packets are located, even if dissipative wave-mean interaction is involved.

Enhancement of GW $F_p$s in the 4D experiment begins about 10 days before the SSW onset date, and it remains several days even after the onset in the recovery phase. Significant increase of the westward $F_p$s related to the wave capture starts 10 days before the onset date in the USLM, and enhancement of the eastward $F_p$ in the lower thermosphere also begins from early stage of the SSW evolution. In the 2D experiment, vertical propagation is quite restrictive, and significant $F_p$ are not found in the lower thermosphere before the SSW onset. In the mesosphere, eastward $F_p$s are substantially enhanced a few days before the onset in the 4D for both OGWs and NGWs, but relatively weak eastward $F_p$s are induced near the onset in the 2D for NGWs alone. In the recovery phase, eastward $F_p$s are also enhanced around $z = 40$ km in the 4D, which implies that recovery of the stratospheric jets would possibly be accelerated when realistic propagation is considered.

Interpretation of results shown in this paper may depend on the gridding method designed to generate gridded model outputs. In this method, the spatial size of GW packets is assumed to be as large as horizontal and vertical grid spacings used in this study. This implicit assumption may lead to overestimation of the magnitude of the GW $F_p$s enhanced by the horizontal refraction, since severe horizontal refraction may stretch significantly anisotropically GW packets. In this case of substantial deformation of GW packets, the packets may not occupy entirely grid spacings. Physically, the size and shape of GW packets are also important, since they may affect how GWs interact with mean flows. As Bühler (2014) described, as GW packets occupy more and more spaces in the longitudinal direction, they can influence more locally the large-scale flows where the packets are located. GW packets confined in limited areas may affect the ambient flows in more nonlocal ways. In order to consider properly size of GW packets in space and time, one may need information about how much GW fields are steadily generated from sources [e.g., A generation time scale $\Delta t_g$ is large (small) for steady (intermittent) sources] and how much GW fields occupy horizontal and vertical spaces from the sources (e.g., $|c_{gh}|\Delta t_g$ and $|c_{gz}|\Delta t_g$).

In the present study we have not discussed about how GW momentum forcing can be estimated from the ray simulation results. As described above, consideration of realistic propagation of localized GW packets in the slowly varying large-scale



flows requires for GWPs to compute influences of GWs in more nonlocal ways in space and time, which violates the basic assumptions of current modeling frameworks. In SSW cases as considered in this study, large-scale flows can vary rapidly in space and time, and the nonlocal approach may particularly be more important, since GWs can change vortex structure located around the GWs. However, at this point, it is not straightforward to present in a clear way how to estimate the nonlocal

influences of GWs. In order to consider the nonlocality in models, one might either somehow extend columnar GWPs or explicitly impliment ray-tracing formulations. One way or the other, further theoretical developments on GW processes seem to be necessary, as long as physically-based methods with minimal ad-hoc treatments are preferred.

.

*Code availability.* The HWM14 and DWM07 model codes (Fortran) are included in the supporting information of Drob et al. (2015).

The NRLMSISE-00 model code (Fortran) is provided by Community Coordinated Modeling Center at NASA GSFC (Hedin, 2001). The source codes of the ODEPACK (Fortran) can be downloaded at Lawrence Livermore National Laboratory (Hindmarsh, 2006). The tricubic interpolation code (C++) is obtained from a GitHub repository (Bigaouette, 2015).

*Data availability.* The ERA-Interim data are obtained using Meteorological Archival and Retrieval System (MARS) of the European Centre for Medium-Range Weather Forecasts (ECMWF, 2009). The MERRA-2 data are obtained through Goddard Earth Sciences Data and

Information Services Center (GES DISC, GMAO, 2015). The NOGAPS-ALPHA data are available at a public domain managed by US Naval Research Laboratory (NRL, 2009). The F10.7 solar flux and geomagnetic *Ap* indices are provided by NOAA National Centers for Environmental Information (NCEI, 2018). Grid- and subgrid-scale topography data are obtained from NCAR CESM inputdata repository (NCAR, 2019)

## Appendix A:  Detailed model description

**A1   Derivation of ray-tracing equations**

Local time change of $\omega$ is obtained by taking the time derivative of Eq. (3) and can be written using Eqs. (1)–(3) as

$$\partial\omega/\partial t = -\partial\Omega/\partial \boldsymbol{k} \cdot \boldsymbol{\nabla_r}\omega + (\partial\Omega/\partial\Lambda_n)\,\partial\Lambda_n/\partial t, \tag{A1}$$

where $\partial\Omega/\partial\boldsymbol{k}$ corresponds to the group velocity $\boldsymbol{c}_g$.

From the definition of $d/dt$ $(=\partial/\partial t + \boldsymbol{c}_g \cdot \boldsymbol{\nabla_r})$, it is clear that Eq. (A1) is the same as Eq. (5). By definition, $\boldsymbol{c}_g = d\boldsymbol{r}/dt$

(where $\boldsymbol{r} = r\boldsymbol{e}_r$). This is proved by substituting $\boldsymbol{r} = r\boldsymbol{e}_r$ into $d\boldsymbol{r}/dt$ and by using $\partial\boldsymbol{r}/\partial t = 0$, $\partial\boldsymbol{r}/\partial\lambda = r\cos\phi\boldsymbol{e}_\lambda$, $\partial\boldsymbol{r}/\partial\phi = r\boldsymbol{e}_\phi$, and $\partial\boldsymbol{r}/\partial r = \boldsymbol{e}_r$ (i.e., $d\boldsymbol{r} = r\cos\phi d\lambda\boldsymbol{e}_\lambda + rd\phi\boldsymbol{e}_\phi + dr\boldsymbol{e}_r$). As a result, a trajectory of a wave packet is described as follows:

$$(r\cos\phi d\lambda/dt, rd\phi/dt, dr/dt) = (c_{g\lambda}, c_{g\phi}, c_{gr}). \tag{A2}$$





Local time change of $\boldsymbol{k}$ is obtained by taking the time derivative of Eq. (2) and is written using Eqs. (1)–(3) as

$$\partial \boldsymbol{k}/\partial t = -\partial \Omega/\partial \boldsymbol{k} \cdot \boldsymbol{\nabla_r} \boldsymbol{k} - \partial \Omega/\partial \Lambda_n \boldsymbol{\nabla_r} \Lambda_n, \tag{A3}$$

where $\partial \Omega/\partial \boldsymbol{k} \cdot \boldsymbol{\nabla_r} \boldsymbol{k}$ is expressed using summation index as $\partial \Omega/\partial k_i \boldsymbol{\nabla_r} k_i$ because the two $\boldsymbol{k}$s contract with each other.

Since $k_i = \boldsymbol{e}_i \cdot \boldsymbol{k}$, $\partial \Omega/\partial k_i \boldsymbol{\nabla_r} k_i = \partial \Omega/\partial k_i \boldsymbol{\nabla_r} (\boldsymbol{e}_i \cdot \boldsymbol{k})$, and thus Eq. (A3) becomes

$$d\boldsymbol{k}/dt = -c_{gi} \left( \boldsymbol{\nabla_r} \boldsymbol{e}_i \right) \cdot \boldsymbol{k} - \partial \Omega/\partial \Lambda_n \boldsymbol{\nabla_r} \Lambda_n. \tag{A4}$$

Here, $c_{gi} \left( \boldsymbol{\nabla_r} \boldsymbol{e}_i \right) \cdot \boldsymbol{k}$ should be zero for invariance with respect to choice of coordinate system (Sect. 2.2). Consequently, Eq. (A4) is reduced to the equation for $\boldsymbol{k}$ in Eq. (4).

The constraint $c_{gi} \left( \boldsymbol{\nabla_r} \boldsymbol{e}_i \right) \cdot \boldsymbol{k} = 0$ indicates that the following two relations should always be satisfied on sphere:

$$kc_{g\phi} \tan \phi + mc_{g\lambda} = lc_{g\lambda} \tan \phi + kc_{gr}, \tag{A5}$$

$$\text{and} \quad lc_{gr} = mc_{g\phi}. \tag{A6}$$

Note that these relations are derived from spatial variations of the basis vectors (i.e., $\boldsymbol{\nabla_r} \boldsymbol{e}_i$).

Substituting $\boldsymbol{k} = k\boldsymbol{e}_\lambda + l\boldsymbol{e}_\phi + m\boldsymbol{e}_r$ into Eq. (A4) [where $c_{gi} \left( \boldsymbol{\nabla_r} \boldsymbol{e}_i \right) \cdot \boldsymbol{k} = 0$] gives component forms of Eq. (A4):

$$dk/dt = -\left( \Omega_n/h_\lambda \right) \partial \Lambda_n/\partial \lambda + C_{k1} + C_{k2}, \tag{A7}$$

$$dl/dt = -\left( \Omega_n/h_\phi \right) \partial \Lambda_n/\partial \phi + C_{l1} + C_{l2}, \tag{A8}$$

$$\text{and} \quad dm/dt = -\left( \Omega_n/h_r \right) \partial \Lambda_n/\partial r + C_{m1} + C_{m2}, \tag{A9}$$

where $\Omega_n = \partial \Omega/\partial \Lambda_n$; $h_\lambda = r \cos \phi$, $h_\phi = r$, and $h_r = 1$; Terms denoted by $C$ represent curvature effects: $C_{k1} = (lc_{g\lambda} \tan \phi)/r$, $C_{k2} = -mc_{g\lambda}/r$, $C_{l1} = -(kc_{g\lambda} \tan \phi)/r$, $C_{l2} = -mc_{g\phi}/r$, $C_{m1} = kc_{g\lambda}/r$, and $C_{m2} = lc_{g\phi}/r$.

From Eqs. (A7)–(A9), it can be shown that the magnitude of a 3D wavenumber vector is invariant with respect to the earth's curvature by multiplying Eq. (A7) by $k$, Eq. (A8) by $l$, and Eq. (A9) by $m$ and by adding these results all together.

In the model, Eqs. (4) and (5) are approximated for the shallow atmosphere, and for the dispersion relation Eq. (6), they can be written in a component form as follows:

$$d\lambda/dt = \left[ U + kN_{\hat{\omega}}^2 / \left( \hat{\omega} \sigma^2 \right) \right]/h_\lambda = c_{g\lambda}/h_\lambda, \tag{A10}$$

$$d\phi/dt = \left[ V + lN_{\hat{\omega}}^2 / \left( \hat{\omega} \sigma^2 \right) \right]/h_\phi = c_{g\phi}/h_\phi, \tag{A11}$$

$$dz/dt = -m\hat{\omega}_f^2 / \left( \hat{\omega} \sigma^2 \right) = c_{gz}, \tag{A12}$$

$$dk/dt = -\left( kU_\lambda + lV_\lambda + M_\lambda \right)/h_\lambda + C_{k1}, \tag{A13}$$

$$dl/dt = -\left( kU_\phi + lV_\phi + M_\phi \right)/h_\phi + C_{l1}, \tag{A14}$$

$$dm/dt = -\left( kU_z + lV_z + M_z \right), \tag{A15}$$

$$\text{and} \quad d\omega/dt = +\left( kU_t + lV_t + M_t \right), \tag{A16}$$





where $N_{\hat{\omega}}^2 = N^2 - \hat{\omega}^2$; $\hat{\omega}_f^2 = \hat{\omega}^2 - f^2$; $\sigma^2 = k^2 + l^2 + m^2 + \alpha^2$; $h_\lambda = a\cos\phi$, and $h_\phi = a$; $U_\lambda$, $U_\phi$, $U_z$, and $U_t$ ($V_\lambda$, $V_\phi$, $V_z$, and $V_t$) denote the partial derivatives of $U$ ($V$) with respect to $\lambda$, $\phi$, $z$, and $t$, respectively; $C_{k1} = (l c_{g\lambda}\tan\phi)/a$; $C_{l1} = -(k c_{g\lambda}\tan\phi)/a$.

In Eqs. (A13)–(A16), terms starting with $M$, effects of background properties other than $U$ and $V$, are given by

$$M_\lambda = 1/(2\hat{\omega}\sigma^2)\left(k_h^2 N_\lambda^2 - \hat{\omega}_f^2 \alpha_\lambda^2\right), \tag{A17}$$

$$M_\phi = 1/(2\hat{\omega}\sigma^2)\left(k_h^2 N_\phi^2 + m_\alpha^2 f_\phi^2 - \hat{\omega}_f^2 \alpha_\phi^2\right), \tag{A18}$$

$$M_z = 1/(2\hat{\omega}\sigma^2)\left(k_h^2 N_z^2 - \hat{\omega}_f^2 \alpha_z^2\right), \tag{A19}$$

and $$M_t = 1/(2\hat{\omega}\sigma^2)\left(k_h^2 N_t^2 - \hat{\omega}_f^2 \alpha_t^2\right), \tag{A20}$$

where $N_\lambda^2$, $N_\phi^2$, $N_z^2$, and $N_t^2$ ($\alpha_\lambda^2$, $\alpha_\phi^2$, $\alpha_z^2$, and $\alpha_t^2$) denote the partial derivatives of $N^2$ ($\alpha^2$) with respect to $\lambda$, $\phi$, $z$, and $t$, respectively; $f_\phi^2 = \partial f^2/\partial\phi$; $k_h^2 = k^2 + l^2$; $m_\alpha^2 = m^2 + \alpha^2$.

Under the shallow-atmosphere approximation (Phillips, 1966) where curvature terms related to vertical movements are ignored, there is no relation corresponding to Eq. (A6), and Eq. (A5) is reduced to

$$k c_{g\phi}\tan\phi = l c_{g\lambda}\tan\phi. \tag{A21}$$

Using Eqs. (A13) and (A14), it can be proved that the magnitude of horizontal wavenumber vector is invariant with respect to curvature effects as in Eqs. (A7)–(A9).

## A2   Effects of viscosity and diffusivity on GWs

Viscous damping and thermal diffusion terms for GWs can be obtained by linearizing the viscosity term (derived from the symmetric stress tensor) in Navier-Stokes equation and the diffusion term in thermodynamic energy equation (see Kundu, 1990; Vadas and Fritts, 2005) as follows:

$$\nu\left[\nabla^2 \boldsymbol{v}' + (1/3)\boldsymbol{\nabla}\left(\boldsymbol{\nabla}\cdot\boldsymbol{v}'\right)\right], \tag{A22}$$

and $$(\nu/\mathrm{Pr})\left(1/\bar{T}\right)\nabla^2 T', \tag{A23}$$

where $\boldsymbol{v}'$ [$= (u', v', w')$] is the 3D perturbation wind vector; $u'$, $v'$, and $w'$ are the zonal, meridional, and vertical perturbation wind components, respectively; $T'$ is the temperature perturbation; $\bar{T}$ is the background temperature; $\nu$ is the kinematic viscosity; Pr is the Prandtl number.

In the viscosity terms Eq. (A22), $(\nu/3)\boldsymbol{\nabla}\left(\boldsymbol{\nabla}\cdot\boldsymbol{v}'\right)$ is ignored by assuming that GW vertical wavelengths are much smaller than $4\pi H$ (Vadas and Fritts, 2005), where $H$ is the density scale height. Diffusivity term Eq. (A23) is reduced to $(\nu/\mathrm{Pr})\nabla^2 b'$ (where $b' = \theta'/\bar{\theta}$, $\theta'$ and $\bar{\theta}$ are the perturbation and background potential temperatures, respectively), by neglecting pressure perturbations and spatiotemporal variations of background variables compared to GW phase variations.

Viscous damping and thermal diffusion may affect propagation of GWs through modification of dispersion relation (Vadas and Fritts, 2005) as well as amplitudes, but in this model effects on amplitudes are only considered. In order to obtain a





closed expression for $\tau_{\text{dis}}$ in Eq. (7), following Marks and Eckermann (1995), the approximated damping terms [$\nu\nabla^2\boldsymbol{v}'$ and $(\nu/\text{Pr})\nabla^2 b'$] are modified, albeit somewhat arbitrarily, to a density-weighted form:

$$K\left[\partial^2\chi'/\partial x^2 + \partial^2\chi'/\partial y^2 + \bar{\rho}^{-1}\partial/\partial z\left(\bar{\rho}\partial\chi'/\partial z\right)\right], \tag{A24}$$

where $\chi'$ is $u'$, $v'$, $w'$, or $b'$; $K$ is either the kinematic viscosity ($\nu$) or the thermal diffusivity ($\nu/\text{Pr}$).

After substituting plane-wave solutions such as $\chi' = e^{z/(2H)}\hat{\chi}e^{i(kx+ly+mz-\omega t)}$ into Eq. (A24), derivation of equations for GW energy and action averaged over phases gives the right-hand side of Eq. (7):

$$-2\nu\sigma^2\left(X + Y_{\text{pr}}\right)/\left(X + Y\right)A, \tag{A25}$$

where $X = (\hat{\omega}^2+f^2)(k^2+l^2)/(\hat{\omega}^2-f^2)^2$, $Y_{\text{pr}} = (\hat{\omega}^2+\text{Pr}^{-1}N^2)(m^2+\alpha^2)/(N^2-\hat{\omega}^2)^2$, and $Y = (\hat{\omega}^2+N^2)(m^2+\alpha^2)/(N^2-\hat{\omega}^2)^2$.

Therefore, $\tau_{\text{dis}}$ becomes

$$\tau_{\text{dis}} = 1/(2\nu\sigma^2)\left(X + Y\right)/\left(X + Y_{\text{pr}}\right). \tag{A26}$$

When $\text{Pr} = 1$, $Y_{\text{pr}} = Y$, and Eq. (A26) is reduced to Eq. (12).

**A3   Details of numerical implementation**

The LSODA solver employs subtime stepping within each $\delta t$. Sub-timestep is determined so that the maximum norm of relative errors can be less than 1. The relative error ($e_r$) of each solution ($y$) is defined by solver-estimated error ($e$) divided by an weight ($w$) ($e_r = e/w$), where $w = t_r|y| + t_a$, and $t_r$ and $t_a$ are relative and absolute tolerances specified for each $y$, respectively. For $\lambda$, $\phi$, and $z$ ($k$, $l$, $m$, and $\omega$), $t_r$ and $t_a$ are specified as $10^{-3}$, and $10^{-6}$ ($10^{-6}$, and $10^{-9}$), respectively. Some sensitivity tests on thresholds are carried out, but threshold values smaller than specified above do not give significantly different results. One example of the sensitivity tests can be found in the supplement (Fig. S3).

In the gridding method, the horizontal projection of a 3D ray trajectory during $\delta t$ is assumed to be represented by a great-circle path, the shortest path between two points on sphere. For an initial location ($\lambda_i$, $\phi_i$, $z_i$), time integration of the ray tracing equations gives a final location ($\lambda_f$, $\phi_f$, $z_f$) after $\delta t$. Spherical arc lengths ($ds$) from the final horizontal position to the centers ($\lambda_c$, $\phi_c$) of 8 horizontal grid cells adjacent to the initial horizontal position are computed using

$$d = \cos^{-1}\left(\sin\phi_c\sin\phi_f + \cos\phi_c\cos\phi_f\cos\delta\lambda\right), \tag{A27}$$

where $\delta\lambda = |\lambda_c - \lambda_f|$. Among the 8 cell-center locations ($\lambda_c$, $\phi_c$), one cell that gives minimum $d$ is chosen, and then identical procedure is repeated for 8 neighboring horizontal grid cells around the chosen cell until a grid cell that contains the final horizontal position is approached. Determination of contiguous 3D grid cells between ($\lambda_i$, $\phi_i$, $z_i$) and ($\lambda_f$, $\phi_f$, $z_f$) is completed considering how many vertical grid cells the ray move through while it pass through the chosen horizontal grid cells.


Using this gridding method, three components of group velocity are stored at the vertices of chosen grid cells between initial and final positions. In addition, various ray properties such as $k$, $l$, $m$, $\omega$, $\hat{\omega}$, $F_A$, and $\boldsymbol{F}_p$ including forcing terms of the ray-tracing equations are stored at the same grid vertices to generate gridded model outputs.

In the model, rays are eliminated when some criteria are satisfied after time integration for $\delta t$: (i) when rays move out of the
5   model atmosphere through top and bottom boundaries, (ii) when rays are 3-day old, (iii) when magnitude of the pseudomomentum flux ($|\boldsymbol{k}_h c_{gz} A|$) is less than $10^{-10}$, or (iv) when time integration results are numerically invalid. In the present model, rays are not eliminated owing to WKB criteria based on the finding (Sartelet, 2003) that ray theory can work remarkably well in spite of the local breakdown of scale separation between GWs and large-scale flow. For rays to be eliminated, the gridding procedure is not carried out, and thus those rays do not affect $\tau_{\mathrm{def}}$ and gridded outputs.

*Author contributions.*  ISS and CL planned this study. ISS developed the ray-tracing model, designed ray-tracing simulations, carried out
10   model experiments, and analyzed model results. HYC provided advices on interaction between gravity waves and planetary waves. BGS provided information about previous studies about the 2009 SSW. All coauthors gave comments on the manuscript. ISS wrote the manuscript.

*Competing interests.*  The authors declare that they have no conflict of interest.

*Acknowledgements.*  This study was supported by research fund PE19020 from Korea Polar Research Institute and funded by the KMA/NMSC's
15   (Korea Meteorological Administration/National Meteorological Satellite Center) project (NMSC-2016-3137). Also, this work was supported by the National Institute of Supercomputing and Network/Korea Institute of Science and Technology Information with supercomputing resources including technical support (KSC-2016-C2-0034). HYC is supported by the National Research Foundation of Korea (NRF) grant funded by the Korea government (MSIT) (No. 2017R1A2B2008025).





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



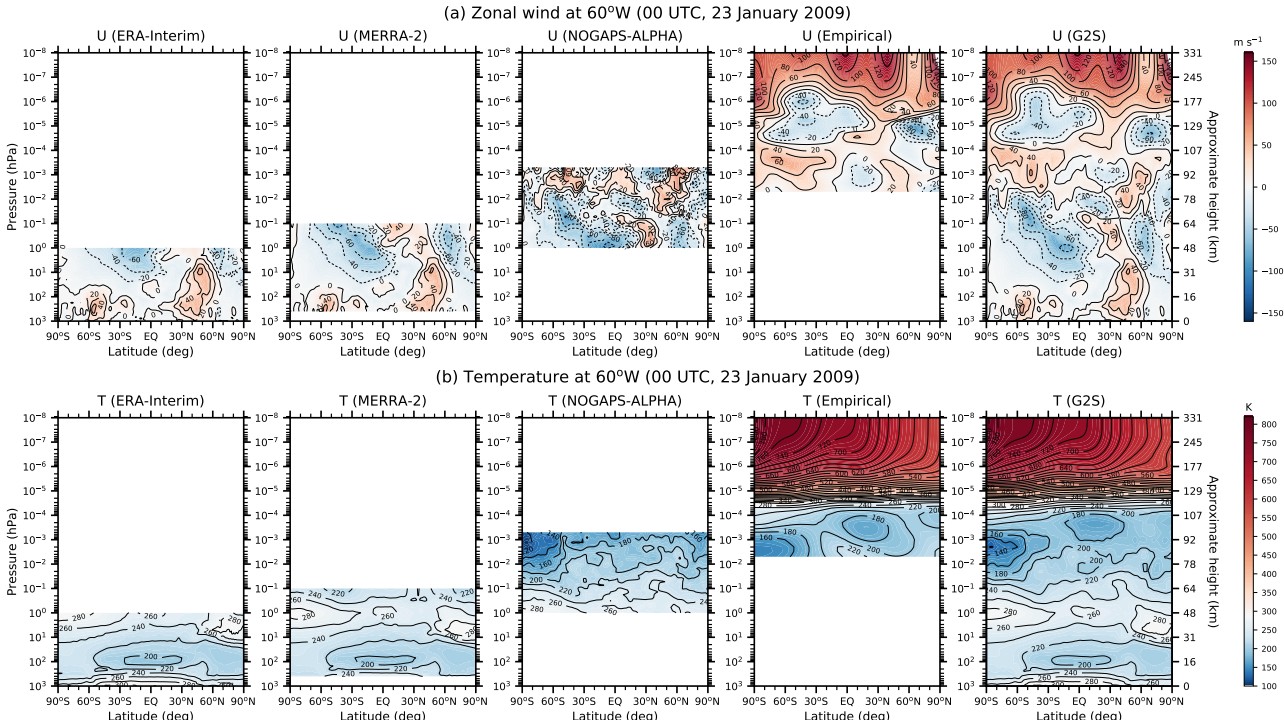

**Figure 1.** Latitude-height cross sections of (a) zonal wind and (b) temperature in the ERA-Interim, MERRA-2, NOGAPS-ALPHA, empirical models, and G2S data at 60°W at 00 UTC on 23 January 2009. For zonal wind, shading and contour intervals are 2 and 20 m/s, respectively. Contours for westward winds are plotted in dotted lines. For temperature, shading and contour intervals are 5 and 20 K, respectively.

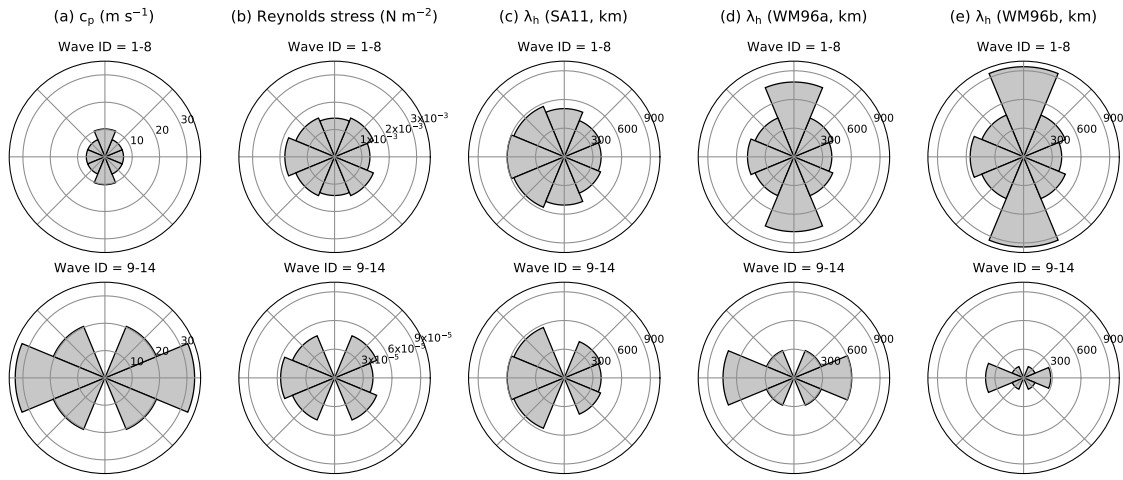

**Figure 2.** Angular histograms of (a) phase speeds and (b) Reynolds stresses, and (c)–(e) horizontal wavelengths of nonorographic GWs (SA11, WM96a, and WM96b) as a function of propagation directions ($\varphi$) at an interval of $45°$. For wave IDs of 1–8 (9–14), $c_p = 6.8, 6.8, 10.2, 6.8, 6.8, 6.8, 10.2,$ and $6.8$ m s$^{-1}$ (32.8, 20.4, 20.4, 32.8, 20.4, and 20.4 m s$^{-1}$) counterclockwise from the due East ($\varphi = 0°$).





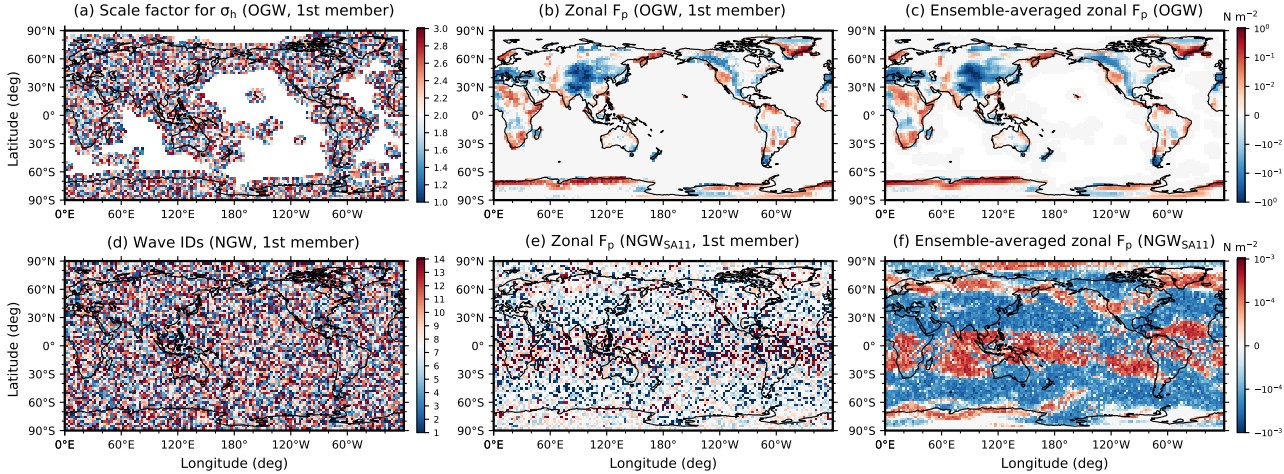

**Figure 3.** Longitude-latitude distributions of [(a) and (d)] stochastic parameters and [(b), (c), (e), and (f)] zonal pseudomomentum fluxes ($F_p$) for OGWs and NGW$_{SA11}$s at 00 UTC on 20 January 2009 on the $2.5° \times 2.5°$ horizontal grid: (a) Scale factors for the standard devation of the subgrid-scale topography for the 1st OGW ensemble member, (b) zonal OGW $F_p$ above source layers for the 1st OGW ensemble member, (c) ensemble-averaged zonal OGW $F_p$, (d) wave IDs for the 1st NGW ensemble member, (e) zonal NGW $F_p$ at $z = 6.8$ km for the 1st NGW ensemble member, and (f) ensemble-averaged zonal NGW $F_p$ at $z = 6.8$ km. OGW $F_p$s are multiplied by an efficiency factor (0.125) as described in Richter et al. (2010).



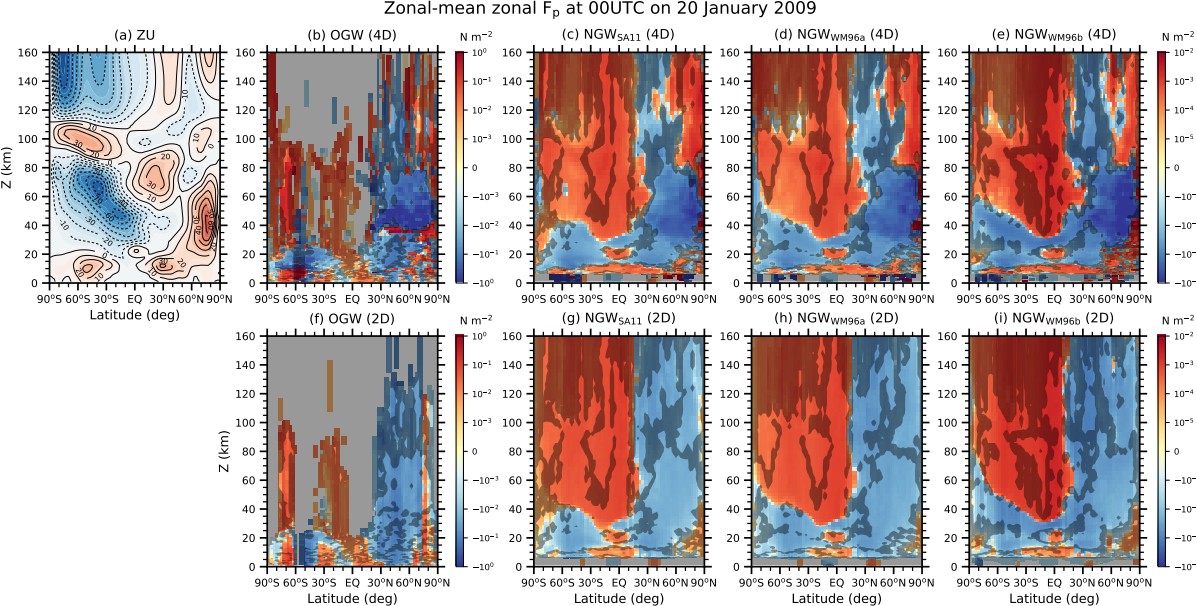

**Figure 4.** Latitude-height cross-sections of (a) zonal-mean zonal wind (ZU) and (b–i) zonal-mean zonal pseudomomentum fluxes ($F_p$s) for OGW and three NGW schemes in the (top) 4D and (bottom) 2D experiments at 00 UTC on 20 January 2009. OGW $F_p$s are multiplied by the efficiency factor (0.125). Contour interval of zonal-mean zonal wind is 10 m s$^{-1}$ and negative values are plotted in dashed lines. Transparently shaded areas on the pseudomomentum fluxes indicate regions where the paired and two-tailed $t$-test for 20 ensemble members of the 4D and 2D experiments gives $p$ values larger than 0.05 (i.e., no statistical significance at the level of 0.05). Here, the $p$ value means probability that mean values in the 4D and 2D experiments would be similar to each other. The shaded area are identical in the pair of 4D and 2D experiments for a particular GW scheme. Nonparameteric test such as Wilcoxon's signed rank test, where no probabilistic distribution is assumed, also gives almost similar results as the $t$-test. For these statistical tests, algorithms presented in Boslaugh (2013) are employed.



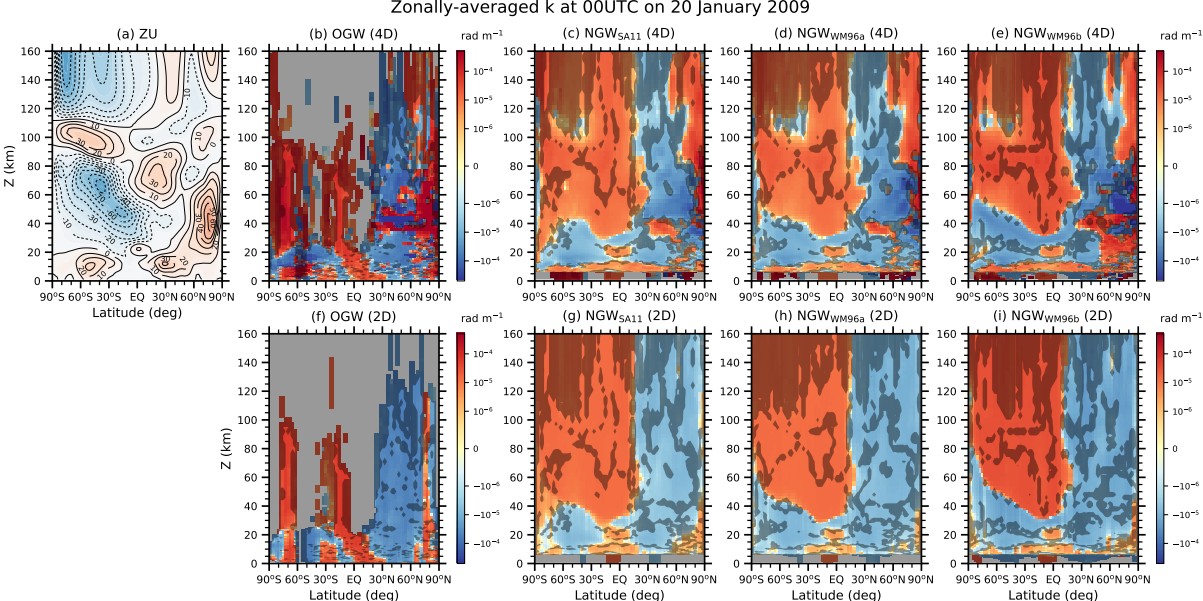

**Figure 5.** Latitude-height cross-sections of (a) zonal-mean zonal wind (ZU) and (b–i) zonal-mean zonal wavenumbers ($k$s) for OGW and three NGW schemes in the 4D and 2D experiments at 00 UTC on 20 January 2009. Contour interval of zonal-mean zonal wind is 10 m s$^{-1}$ and negative values are plotted in dashed lines. Transparently shaded areas on the zonal wavenumbers indicate regions where the paired and two-tailed $t$-test for the 4D and 2D experiments gives $p$ values larger than 0.05 (i.e., no statistical significance at the level of 0.05).



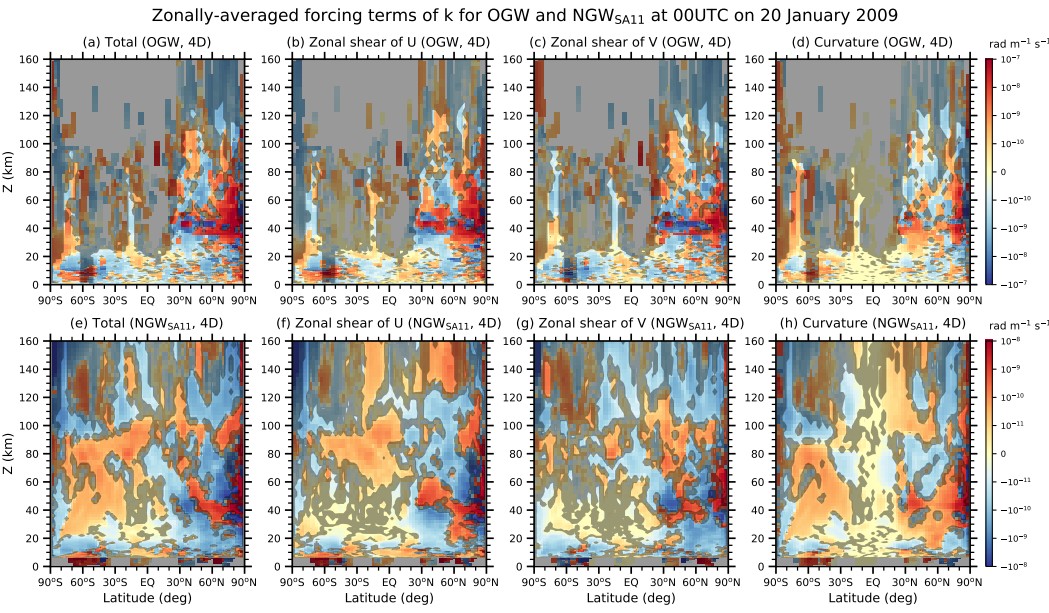

**Figure 6.** Latitude-height cross-sections of total and three major forcing terms [The zonal shear terms of the large-scale zonal and meridional winds ($U$ and $V$) and the curvature term] of the zonal wavenumber for (top) OGWs and (bottom) NGW$_{SA11}$s in the 4D experiment at 00 UTC on 20 January 2009. The zonal shear terms of $U$ and $V$ are $-k/(a\cos\phi)\partial U/\partial\lambda$ and $-l/(a\cos\phi)\partial V/\partial\lambda$, respectively. The curvature term is given by $lc_{g\lambda}\tan\phi/a$. Transparently shaded areas indicate regions where the paired and two-tailed $t$-test for the 4D and 2D experiments gives $p$ values larger than 0.05 (i.e., no statistical significance at the level of 0.05).

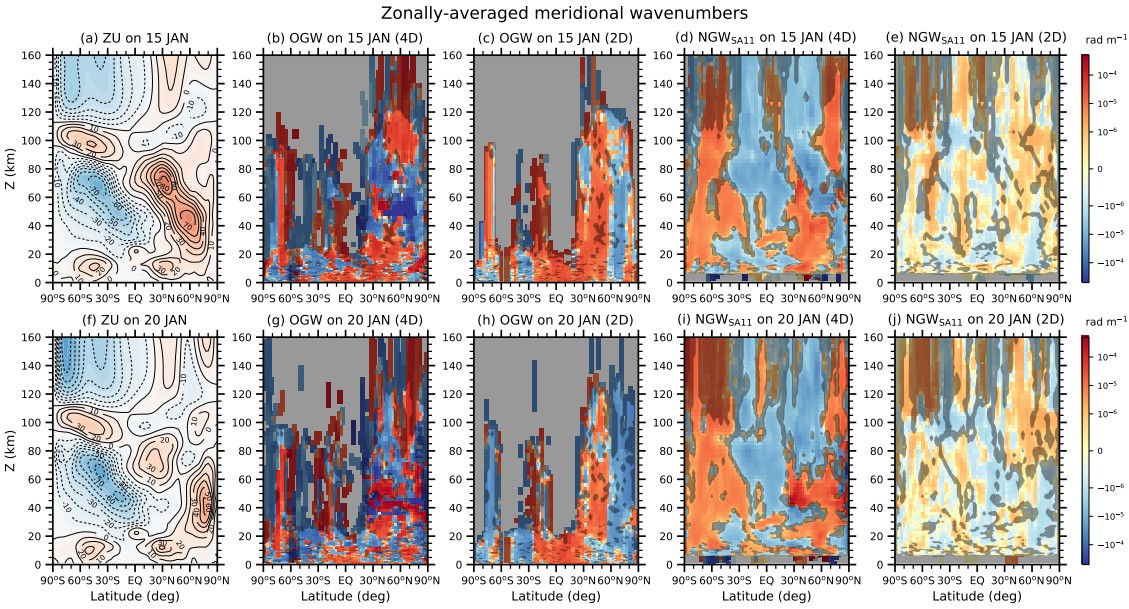

**Figure 7.** Latitude-height cross-sections of zonal-mean zonal wind (ZU) and ensemble means of zonally-averaged meridional wavenumbers ($l$s) for OGWs and NGW$_{SA11}$s in the 4D and 2D experiments at 00 UTC on (top) 15 January and (bottom) 20 January, 2009. Contour interval for zonal-mean zonal wind is 10 m s$^{-1}$ and negative values are plotted in dashed lines. Transparently shaded areas on the meridional wavenumbers indicate regions where the paired and two-tailed $t$-test for the 4D and 2D experiments gives $p$ values larger than 0.05 (i.e., no statistical significance at the level of 0.05).



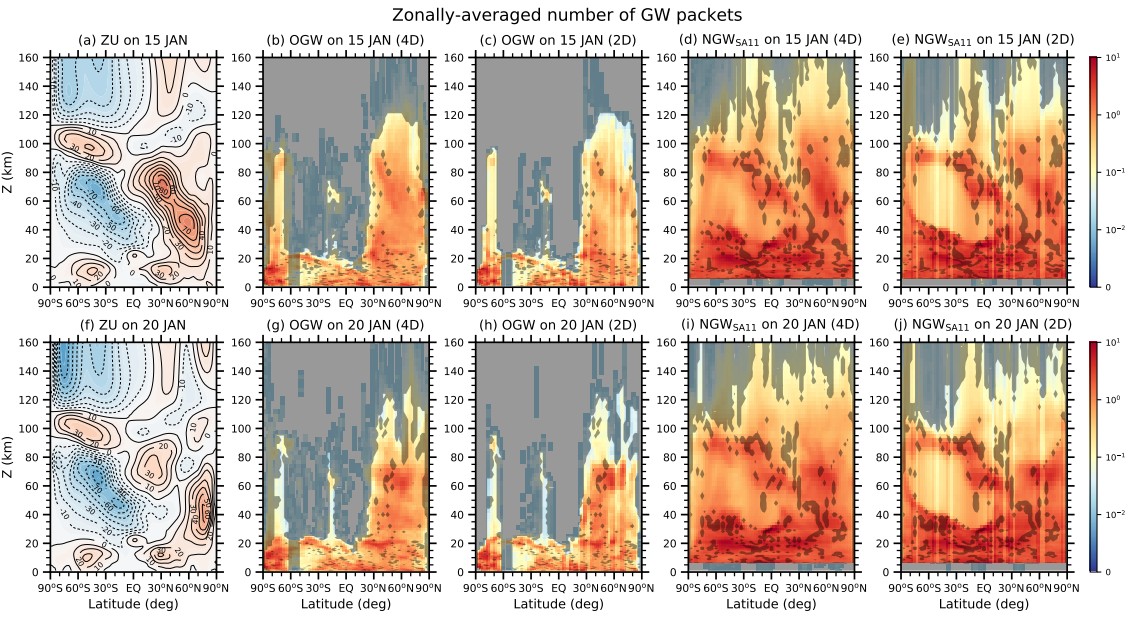

**Figure 8.** Same as Fig. 7 except for zonally-averaged number of GW packets.



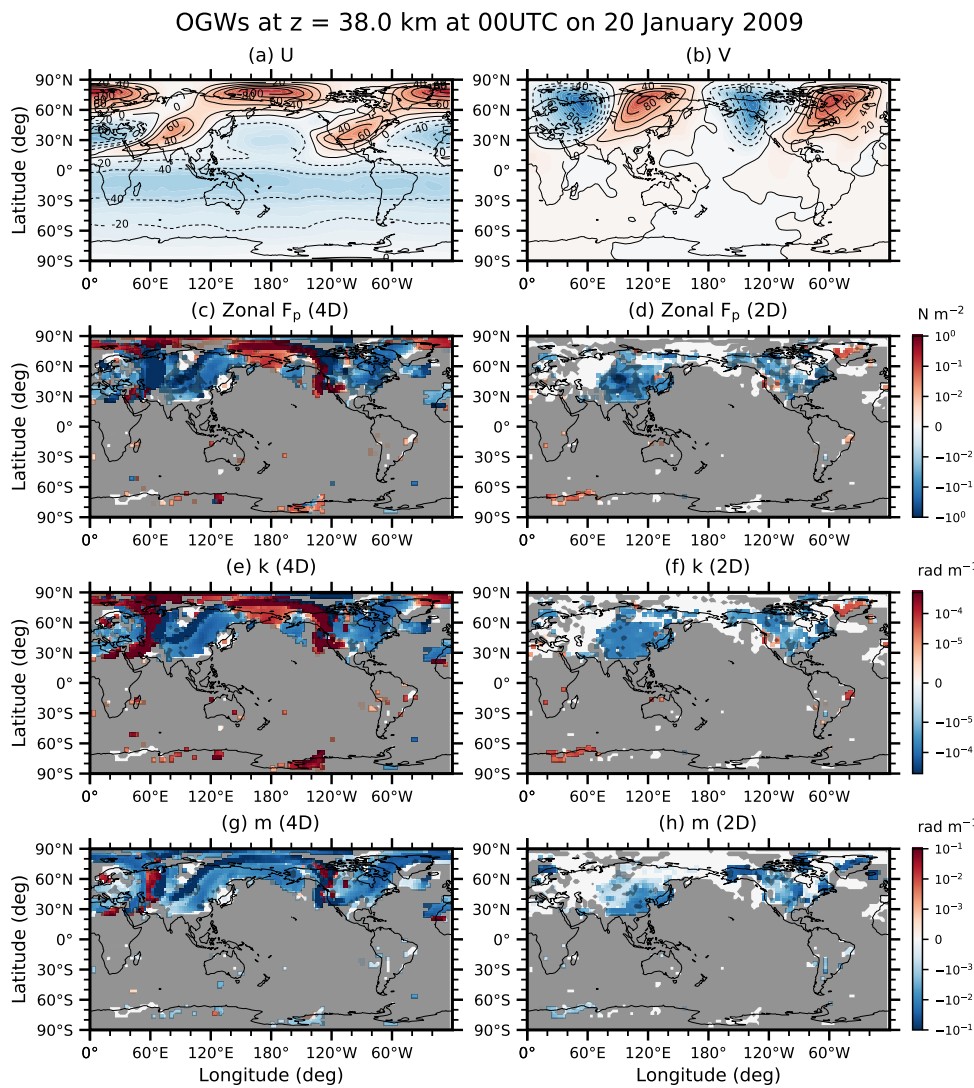

**Figure 9.** Longitude-latitude cross-sections of (a) zonal wind ($U$), (b) meridional wind ($V$), and ensemble means of (c–d) zonal pseudomomentum fluxes ($F_p$s), (e–f) zonal wavenumbers and (g–h) vertical wavenumbers for OGWs at $z = 38$ km in the 4D and 2D experiments at 00 UTC on 20 January, 2009. OGW $F_p$s are multiplied by the efficiency factor (0.125). Contour interval for zonal and meridional winds is 20 m s$^{-1}$ and negative values are plotted in dashed lines. Transparently shaded areas indicate regions where the paired and two-tailed $t$-test for the 4D and 2D experiments gives $p$ values larger than 0.05 (i.e., no statistical significance at the level of 0.05).



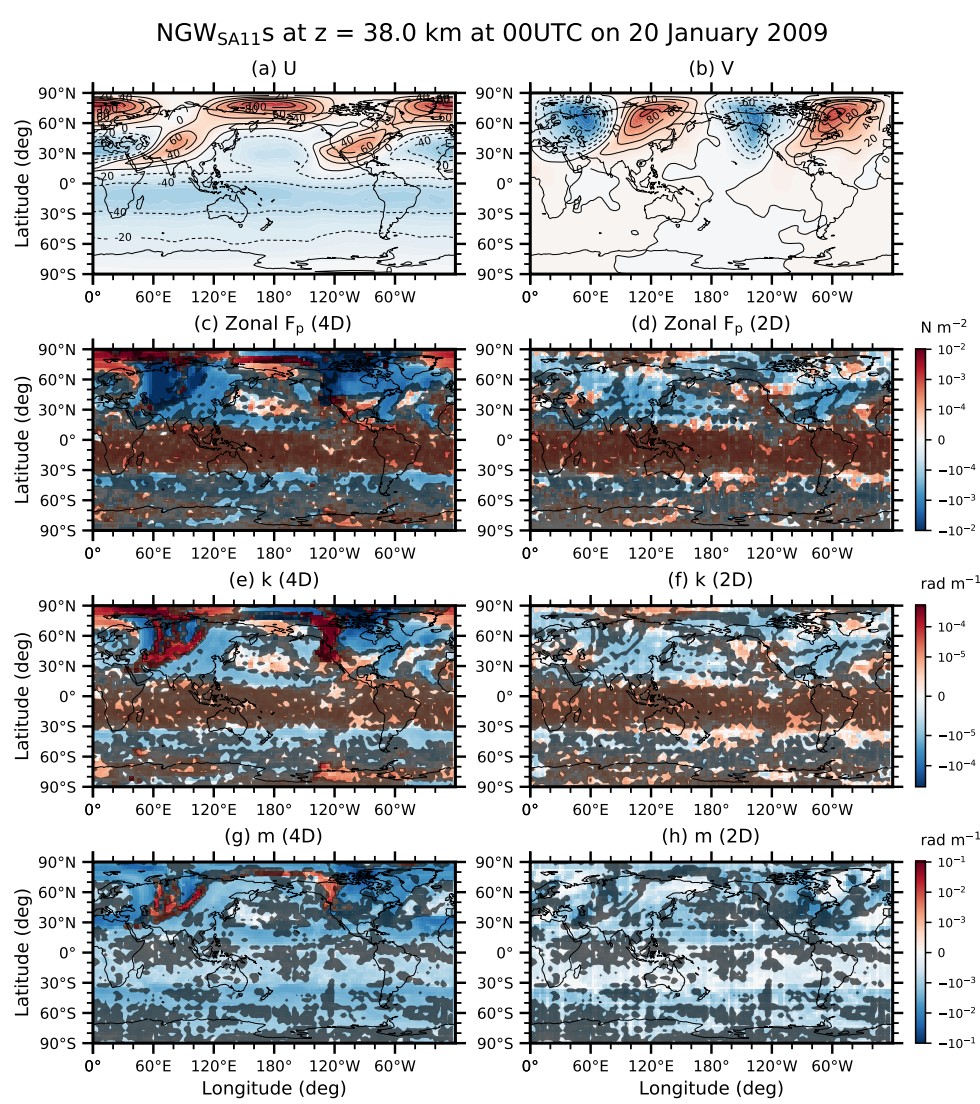

**Figure 10.** Same as Fig. 9 except for NGW$_{SA11}$s.



**Figure 11.** Longitude-latitude cross-sections of (a) zonal wind ($U$), (b) meridional wind ($V$), and ensemble means of the zonal and meridional components of (c–d) the group velocity ($c_{g\lambda}$ and $c_{g\phi}$) and (e–f) the intrinsic group velocity ($c_{g\lambda} - U$ and $c_{g\phi} - V$), (g) the vertical component of the group velocity ($c_{gz}$), and (f) ratio of intrinsic frequency ($\hat{\omega}$) to Coriolis parameter ($|f|$) for OGWs at $z = 38$ km in the 4D experiment at 00 UTC on 20 January, 2009. Contour interval for zonal and meridional winds and horizontal components of group velocities is 20 m s$^{-1}$ and negative values are plotted in dashed lines. In panels (e) and (f), contours of 0 and $\pm 5$ m s$^{-1}$ are only plotted. Transparently shaded areas indicate regions where the paired and two-tailed $t$-test for the 4D and 2D experiments gives $p$ values larger than 0.05 (i.e., no statistical significance at the level of 0.05).

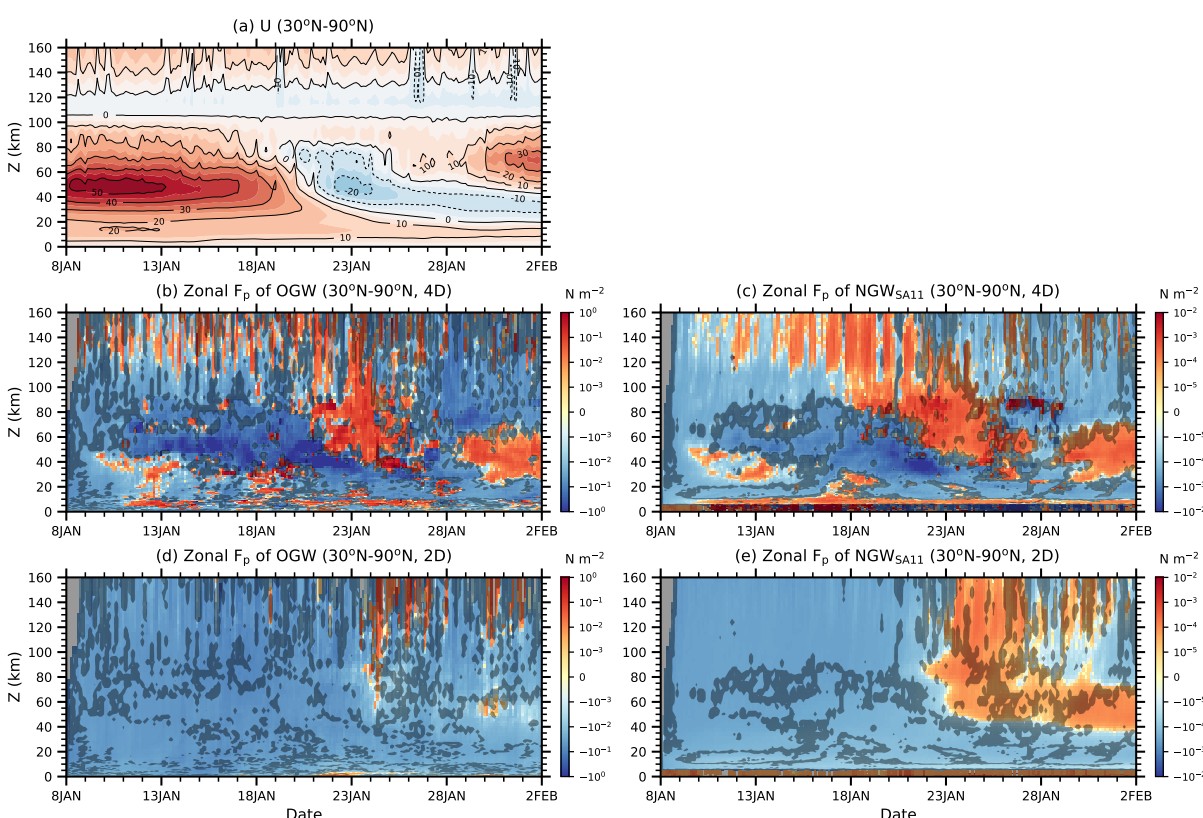

**Figure 12.** Time-height cross-sections of (a) zonal wind ($U$), and (b–e) ensemble means of zonal pseudomomentum fluxes ($F_p$s) averaged over $30°N$–$90°N$ for OGWs and NGW$_{SA11}$s in the (top) 4D and (bottom) 2D experiments. OGW $F_p$s are multiplied by the efficiency factor (0.125). Contour interval for zonal winds is $10~m~s^{-1}$ and negative values are plotted in dashed lines. Transparently shaded areas over the zonal $F_p$s indicate regions where the paired and two-tailed $t$-test for the 4D and 2D experiments gives $p$ values larger than 0.05 (i.e., no statistical significance at the level of 0.05).