# Peer review of "Propagation of gravity waves and its effects on pseudomomentum flux in a sudden stratospheric warming event"

_Atmospheric Chemistry and Physics, 2019_

## Referee Comment (RC1) · Anonymous Referee #2 · 29 Jan 2020

**Review of** "Propagation of gravity waves and its effects on pseudo-momentum flux in a sudden stratospheric warming event"

By In-Sun Song et al.

**Recommendation**: Accept with minor revisions

This is a nicely done, comprehensive study of gravity wave (GW) propagation from tropospheric sources into the middle atmosphere under background wind conditions prevailing during the sudden stratospheric warming of 2009. The authors use a ray tracing model to show how spatial inhomogeneity and evolution of the background flow alters the characteristics of propagating GW; and how this 4D (x, y, z; t) propagation model differs from 2D (z; t) propagation in many nontrivial ways.

The paper is acceptable for publication essentially in its present form. Specific comments below should be addressed to clarify certain points and correct minor errors of grammar and syntax.

**Specific Comments (page, line)**

(1, 16) "may have profound impacts":  Why "may"? GW are the main component of the eddy momentum budget in the mesosphere and above. I would have written "have profound impacts".

(1, 24) "radiatively-driven latitudinal temperature gradient across the two poles"→ pole-to-pole radiatively-driven latitudinal temperature gradient".

(2,1) "Irreversible heat and heat fluxes": This does not make sense. I believe you mean to say "irreversible heat and *momentum* flux *divergences*".

(3, 1) "predominance of non-dissipative wave-mean interaction": "Predominance" overstates the case. Kruse and Smith (2018) stated (their abstract) that "Non-dissipative accelerations are non-negligible and influence a [mountain wave's] approach to breaking, *but breaking and dissipative decelerations quickly develop and dominate the subsequent evolution*" (my italics). Perhaps you meant to say "importance" or "relevance" of non-dissipative interactions? In any case, irreversible changes of the background flow ultimately occur only through dissipation.

(3, 18) "GW activities" → GW activity (this is the standard usage, "activity" here being used as a collective noun).

(4, 19) "where $\Lambda_n$s (n = 1, … N) denote": This is awkward and confusing because the trailing "s", which I believe is intended to denote a plural, could be taken to be part of the symbol.  The standard usage for mathematical symbols is that they do not normally takes an "s" to denote plural. Replace this with "where $\Lambda_n$ (n = 1, … N) denote…"  Note that this occurs many other times through the paper when referring to $\Lambda_n$ and other symbols. Please do a thorough check.

(7, 19) "Then, $\tau_{def}$s are computed" → Then, $\tau_{def}$ is computed. See previous comment. In standard usage $\tau_{def}$ stands for all cases of the "deformation" time scale. No trailing "s" needed.

(10, 4) Figure 3: I would delete panels (a) and (d) of this figure, which do not contain any information that cannot be succinctly explained in the text. On the other hand, there could be a little more discussion of the interesting panels (b)-(c) and (e)-(f). In particular, panel (b) indicates that OGW flux, $F_p$, is well organized in space in a single ensemble member. I presume this is due to the fact that $F_p$ is strongly constrained by the OGW source parameterization, which depends explicitly on orography and low-level wind. By contrast, organization of $F_p$ for the NOGW case only emerges in the ensemble because any single ensemble member is completely stochastic (panels (e) vs. (f)).

(10, 32) "but being weakened" → but is much weakened.

(10, 32) "Transparently shaded areas": This is confusing. "Transparent" implies no shading at all. I believe you are referring to the areas overlain by gray(ish) shading. If so, please explain more clearly. Better yet would be to use some other means (e.g., cross-hatching) to denote the regions of non-significant differences to avoid confusion with the color shading meant to denote flux magnitude/sign.

(11, 4): "westward $F_p$s in the 4D are about 10 (28) times enhanced…": This sentence is nearly incomprehensible. Please break it up into two digestible parts, the first referring to the 10X difference between 4D and 2D models in all but one of the parameterizations; and the second referring to the 28X difference in the case of the WM96b non-orographic parameterization. Also, omit the "s" at the end of $F_p$, here and in many other instances; see comment (4, 19).

(11, 26) "zonal-mean $k$s": Here and elsewhere this should be "zonal-mean $k$"; see comment (4, 19).

(12, 6) "thermodynamic forcing terms": What are these? Are you referring to the dependence on $N$?

(12, 29) "meridional wavenumbers ($l$s)" → Here and elsewhere, "$l$s" should be replaced simply by "$l$"; see comment (4, 19).

(13, 19) Figure 8: This figure shows the striking difference between the 4D and 2D models, especially in the discontinuous (in latitude) appearance of OGW $F_p$. This is a common problem in comprehensive global models, which usually employ 2D columnar GW parameterizations. Although wave-mean flow interaction will tend to reduce these effects, this does not necessarily happen for the right reasons; see discussion about "compensation" of parameterized vs. resolved wave forcing in Cohen et al. (JAS 2013, 2014). It might be worth mentioning this problem.

(15, 19) Figure 11: You might consider showing panels (a) through (f) in vector form (vector background wind, **U**, vector horizontal group velocity, **c**$_g$). This would show more clearly the relationship between **U** and **c**$_g$; and also (for the intrinsic group velocity) the regions where that vector is non-negligible.

(17, 12) "These enhanced eastward $F_p$s, if they exist, may induce more rapid recovery of the stratospheric jets, accelerating downward movement of the ES": This is an interesting effect, which would not be captured by the 2D columnar parameterizations used in most comprehensive models.  Note again that "$F_p$s" should be simply "$F_p$" (no trailing "s").

---

## Referee Comment (RC2) · Anonymous Referee #1 · 12 Feb 2020

Review of manuscript: "Propagation of gravity waves and its effects on pseudomomentum flux in a sudden stratospheric warming event", by In-Sun Song, Changsup Lee, Hye-Yeong Chun, Jeong-Han Kim, Geonhwa Jee, Byeong-Gwon Song, and Julio T. Bacmeister.

This paper investigates the effect of four-dimensional propagation of gravity waves in time-varying background winds on their properties (pseudomomentum fluxes, wavenumber) during the occurrence of a sudden stratospheric warming. The main motivation is that GW parameterizations implemented in climate models generally neglect these effects (columnar, instantaneous propagation is generally enforced in the

GW schemes), and it is important to assess the missing effects on the redistribution of momentum flux and GW forcing. The authors do not find a big difference between 4D and 2D propagation in terms of latitude-height structure of the total momentum fluxes, but do find a significant difference in terms of the magnitude of the momentum fluxes, with much larger fluxes in the 4D scheme. The effects of curvature on the magnitude of the fluxes seems to be as important as the effect of horizontal wind shear.

The study is well-written and easy to follow, and the results are relevant and timing, aligning with current efforts to better understand GW processes in order to improve their parameterizations in climate models. I have a few, very minor comments, and I believe the paper is basically publishable as is.

Comments

1. Some parts of the introduction seem a succession of references, and sometimes it is difficult to follow/understand the line of argument (e.g., paragraphs in page 2).

2. Page 3, line 8-9. Richter et al (2010) attributed the improvement in the SSW frequency in WACCM to the turbulent mountain stress parameterization (which improves near-surface winds and planetary wave generation), not to the source-based nonorographic GW scheme.

3. Section 3. Why do the authors use both ERA-Interim and MERRA fields, if they basically cover the same altitude range? Why not just one reanalysis?

4. Page 10 line 11-12. "Zonal F p s in each OGW ensemble member have locally substantial deviations from the ensemble mean (Fig. 3c) in the major mountain areas". This may be true, but it is not discernible in the figure.

5. Figure 8. I may be missing something, but how is it possible that the number of GW packets increase with height in the 2D simulation? If I understand correctly, in the 2D case the only process adding wave packets to a given column is wave generation at the source level.

---

## Referee Comment (RC3) · Anonymous Referee #3 · 28 Feb 2020

***Journal manuscript review of***:
*"Propagation of gravity waves and its effect on pseudomomentum flux in a sudden stratospheric warming event"*
*By: In-Sun Song et al. 2020*

Overall this is a nice study that puts together a really unique and well-designed set of experiments. The math is explained very clearly and completely and the paper provides a well-documented citation list. Moreover, just getting the various GWD schemes running in WACCM is a noteworthy achievement. In terms of the application of the new GWD packages to a problem, I am excited to see the topic of the effects of GWD on SSWs see more attention and the inclusion of orographic and nonorographic schemes in a sophisticated model setting like WACCM offers the possibility to explore some noteworthy questions.

That said, I do feel like some of those noteworthy questions were not addressed and I think that this is a real missed opportunity. In the text below, I suggest a few ideas (which would require a few additional figures) that I think would be very worth the effort to include. I suggest these ideas because as it stands, this paper does not really discuss the mechanistic effects of GWD on SSWs, rather it simply provides some momentum budgets. Not that providing momentum budgets is not interesting, I just really think that a few additional figures could turn this paper into a something of very high value to the community. To be clear, my acceptance of this paper is not contingent on the authors adding my suggestions, I am simply trying to help improve the relevance of the paper.

**Major comments:**

There are two overarching concepts that I think would make compelling additions to your paper. One involves the effects of GWD on the pre-warming evolution of the vortex (i.e., preconditioning) and the second involves the possibility that GWD increases or decreases the probability of SSW occurrence. For each of these topics, I suggest two figures from the current literature that would provide a good starting point for figures to provide in the current manuscript.

Topic one – preconditioning: Your paper only shows figures for Jan. 20, but one could argue that it is the overall vortex evolution from Jan. 10-20 that is of prime interest in understanding the triggering of this particular SSW. Indeed Figs. 6-10 of Albers and Birner (JAS 2014) show that this period was of notable interest in the development of the SSW and in particular for GWD, it is the zonally asymmetric momentum fluxes that may play an important role in SSW development. Thus can you provide a few additional figures that show the differences in the vortex evolution and zonally asymmetric momentum fluxes for Jan. 10-20? In particular, zonally asymmetric views with GW momentum fluxes and geopotential height contours to indicate vortex shape (as in Figs. 6 and 7 of Albers and Birner) would be very interesting for the various cases that you have run.

Topic two – probability of SSW occurrence: I'm not sure that my second suggestion is possible to accomplish with your current ensemble setup, but in case it is possible, I think

it would provide a very interesting result. In de la Camara et al. (JAS 2017), it was shown that perturbations to the vortex prior to a SSW can cause the vortex to evolve in very different ways. In particular, Figs. 2b, 6b, and 7 provide a very interesting way of seeing how perturbations to the vortex can disrupt vortex evolution, and in some cases, even disrupting the SSW from occurring at all. Now, I realize that your ensembles start about two weeks before the SSW central date, which means that most (all?) of your ensembles have a SSW, but even so, are there systematic differences in how the vortex evolves for the different model setups? Are there ensembles where a SSW does not occur or just barely occurs? In a similar fashion to what I suggest above, I would be particularly interested in seeing figures similar to de la Camara et al. Fig.7 (which is itself similar in character to Figs. 6 and 7 of Albers and Birner); that is, how does a stereographic view of the vortex evolution look between the various model setups?

**Minor comments:**

Figure 9, 10 and S7: These figures are quite difficult to read. Since you are really only concerned with the NH, why not truncate the figures to include on the NH, or perhaps even just 30-90 N?

**References:**

Albers, J. R. and T. Birner (2014): Vortex preconditioning due to planetary and gravity waves prior to sudden stratospheric warmings. J. Atmos. Sci., 71, 4028–4054, doi:10.1175/JAS-D-14-0026.1.

de la Cámara, A., J. R. Albers, T. Birner, R. R. Garcia, P. Hitchcock, D. E. Kinnison, and A. K. Smith, 2017: Sensitivity of sudden stratospheric warmings to previous stratospheric conditions. J. Atmos. Sci., 74, 2857–2877, https:// doi.org/10.1175/JAS-D-17-0136.1.

---

## Author Comment (AC1) · 28 Apr 2020

**General comments**

**Recommendation**: Accept with minor revisions

This is a nicely done, comprehensive study of gravity wave (GW) propagation from tropospheric sources into the middle atmosphere under background wind conditions prevailing during the sudden stratospheric warming of 2009. The authors use a ray tracing model to show how spatial inhomogeneity and evolution of the background flow alters the characteristics of propagating GW; and how

this 4D ($x$, $y$, $z$; $t$) propagation model differs from 2D ($z$; $t$) propagation in many nontrivial ways.

The study is well-written and easy to follow, and the results are relevant and timing, aligning with current efforts to better understand GW processes in order to improve their parameterizations in climate models. I have a few, very minor comments, and I believe the paper is basically publishable as is.

■ Authors would like to thank the reviewer for carefully reading and evaluating the original manuscript. We think we have corrected faithfully our original manuscript according to reviewer's comments. Please refer to the track-change version of revised manuscript for figure, page and line numbers to be mentioned below.

**Specific comments (page, line)**

1. (1, 16) "may have profound impacts": Why "may"? GW are the main component of the eddy momentum budget in the mesosphere and above. I would have written "have profound impacts".

   • "may" is deleted in the line 19 on the page 1 of the track-change version of the revised manuscript.

2. (1, 24) "radiatively-driven latitudinal temperature gradient across the two poles" → "pole-to-pole radiatively-driven latitudinal temperature gradient".

   • The phrase is modified in the line 4 on the page 2 of the track-change version of the revised manuscript following the reviewer's suggestion.

3. (2, 1) "Irreversible heat and heat fluxes": This does not make sense. I believe you mean to say "irreversible heat and momentum flux divergences".

- Modified sentences can be found in the lines 5–7 on the page 2 of the track-change version of the revised manuscript.

4. (3, 1) "predominance of non-dissipative wave-mean interaction": "Predominance" overstates the case. Kruse and Smith (2018) stated (their abstract) that "Non-dissipative accelerations are non-negligible and influence a [mountain wave's] approach to breaking, *but breaking and dissipative decelerations quickly develop and dominate the subsequent evolution*" (my italics). Perhaps you meant to say "importance" or "relevance" of non-dissipative interactions? In any case, irreversible changes of the background flow ultimately occur only through dissipation.

   - Modified sentences can be found in the lines 6–12 on the page 3 in the track-change version of the revised manuscript.

5. (3, 18) "GW activities" → GW activity (this is the standard usage, "activity" here being used as a collective noun).

   - Modifications can be found in several places on the pages 3 and 21 of the track-change version of the revised manuscript.

6. (4, 19) "where $\Lambda_n$s ($n = 1, \cdots, N$) denote": This is awkward and confusing because the trailing "s", which I believe is intended to denote a plural, could be taken to be part of the symbol. The standard usage for mathematical symbols is that they do not normally takes an "s" to denote plural. Replace this with "where $\Lambda_n$ ($n = 1, \cdots, N$) denote ...". Note that this occurs many other times through the paper when referring to $\Lambda_n$ and other symbols. Please do a thorough check.

   - $\Lambda_n$s is changed into $\Lambda_n$ in the line 30 on the page 4 of the track-change version of the revised manuscript.

7. (7, 19) "Then, $\tau_{def}$s are computed" → Then, $\tau_{def}$ is computed. See previous comment. In standard usage $\tau_{def}$ stands for all cases of the "deformation" time scale. No trailing "s" needed.

- The trailing "s" is removed in the line 28 on the page 7 of the track-change version of the revised manuscript.

8. (10, 4) Figure 3: I would delete panels (a) and (d) of this figure, which do not contain any information that cannot be succinctly explained in the text. On the other hand, there could be a little more discussion of the interesting panels (b)-(c) and (e)-(f). In particular, panel (b) indicates that OGW flux, $F_p$, is well organized in space in a single ensemble member. I presume this is due to the fact that $F_p$ is strongly constrained by the OGW source parameterization, which depends explicitly on orography and low-level wind. By contrast, organization of $F_p$ for the NOGW case only emerges in the ensemble because any single ensemble member is completely stochastic (panels (e) vs. (f)).

- Following the reviewer's suggestion, Fig. 3 is modified, and thus panels for the stochastic parameters are removed (see the page 35 of the track-change version of the revised manuscript). Also, more discussions are added at the end of the page 10 and at the beginning of the page 11 of the track-change version of the revised manuscript.

9. (10, 32) "but being weakened" → but is much weakened.

- The phase is changed as suggested in the line 17 on the page 11 of the track-change version of the revised manuscript.

10. (10, 32) "Transparently shaded areas": This is confusing. "Transparent" implies no shading at all. I believe you are referring to the areas overlain by gray(ish)

shading. If so, please explain more clearly. Better yet would be to use some other means (e.g., cross-hatching) to denote the regions of non-significant differences to avoid confusion with the color shading meant to denote flux magnitude/sign.

- Following the reviewer's suggestion, the shaded areas are replaced with hatched areas in all the figures that contained the "transparently shaded regions". Also, "transparently shaded" is changed into "hatched" everywhere in the main text of the revised manuscript.

11. (11, 4): "westward Fps in the 4D are about 10 (28) times enhanced ...": This sentence is nearly incomprehensible. Please break it up into two digestible parts, the first referring to the 10X difference between 4D and 2D models in all but one of the parameterizations; and the second referring to the 28X difference in the case of the WM96b non-orographic parameterization. Also, omit the "s" at the end of $F_p$, here and in many other instances; see comment (4, 19).

- The sentence is broken into two around the lines 21–24 on the page 11 of the track-change version of the revised manuscript.

12. (11, 26) "zonal-mean ks": Here and elsewhere this should be "zonal-mean k"; see comment (4, 19).

- $k$s is replaced with $k$ everywhere as well as in the line 13 on the page 12 of the track-change version of the revised manuscript.

13. (12, 6) "thermodynamic forcing terms": What are these? Are you referring to the dependence on N?

- "thermodynamic forcing terms" is removed, and some explanations are added in the line 28–29 on the page 12 of the track-change version of the revised manuscript.

14. (12, 29) "meridional wavenumbers (ls)" → Here and elsewhere, "ls" should be replaced simply by "l"; see comment (4, 19).

   • "$l$s" is replaced by "$l$" in the line 18 on the page 13 of the track-change version of the revised manuscript. Plural from of mathematical symbols is modified everywhere in the revised manuscript following the reviewer's comment.

15. (13, 19) Figure 8: This figure shows the striking difference between the 4D and 2D models, especially in the discontinuous (in latitude) appearance of OGW $F_p$. This is a common problem in comprehensive global models, which usually employ 2D columnar GW parameterizations. Although wave-mean flow interaction will tend to reduce these effects, this does not necessarily happen for the right reasons; see discussion about "compensation" of parameterized vs. resolved wave forcing in Cohen et al. (JAS 2013, 2014). It might be worth mentioning this problem.

   • Following the reviewer's comments, some discussions are added in the lines 9–20 on the page 15 and the lines 12–14 on the page 21 of the track-change version of the revised manuscript.

16. (15, 19) Figure 11: You might consider showing panels (a) through (f) in vector form (vector background wind, $U$, vector horizontal group velocity, $c_g$). This would show more clearly the relationship between $U$ and $c_g$; and also (for the intrinsic group velocity) the regions where that vector is non-negligible.

   • Following the reviewer's comments, some panels on Fig. 11 are replotted in vector field format. Please see the page 43 of the track-change version of the revised manuscript. Related discussions are rewritten for clarification on the page 17 of the track-change version of the revised manuscript.

17. (17, 12) "These enhanced eastward $F_p$s, if they exist, may induce more rapid recovery of the stratospheric jets, accelerating downward movement of the ES": This is an interesting effect, which would not be captured by the 2D columnar parameterizations used in most comprehensive models. Note again that "$F_p$s" should be simply "$F_p$" (no trailing "s").

- Please see the lines 28–29 on the page 18 of the track-change version of the revised manuscript. Again, the plural form of mathematical sysbols is modified everywhere in the revised manuscript as the reviewer suggested.

---

## Author Comment (AC2) · 28 Apr 2020

**General comments**

This paper investigates the effect of four-dimensional propagation of gravity waves in time-varying background winds on their properties (pseudomomentum fluxes, wavenumber) during the occurrence of a sudden stratospheric warming. The main motivation is that GW parameterizations implemented in climate models generally neglect these effects (columnar, instantaneous propagation is generally enforced in the GW schemes), and it is important to assess the missing

effects on the redistribution of momentum flux and GW forcing. The authors do not find a big difference between 4D and 2D propagation in terms of latitude-height structure of the total momentum fluxes, but do find a significant difference in terms of the magnitude of the momentum fluxes, with much larger fluxes in the 4D scheme. The effects of curvature on the magnitude of the fluxes seems to be as important as the effect of horizontal wind shear.

The study is well-written and easy to follow, and the results are relevant and timing, aligning with current efforts to better understand GW processes in order to improve their parameterizations in climate models. I have a few, very minor comments, and I believe the paper is basically publishable as is.

■ Authors would like to thank the reviewer for reading and evaluating the original manuscript. We have corrected our manuscript according to the reviewer's comments. Please refer to the track-change version of our revised manuscript for figure and line numbers to be mentioned below.

**Minor comments**

1. Some parts of the introduction seem a succession of references, and sometimes it is difficult to follow/understand the line of argument (e.g., paragraphs in page 2).

   • Following reviewer's comment, some redundant references are excluded and the Introduction is somewhat reduced (see pages 2–3 in the track-change version of the revised manuscript).

2. Page 3, line 8-9. Richter et al (2010) attributed the improvement in the SSW frequency in WACCM to the turbulent mountain stress parameterization (which

improves near-surface winds and planetary wave generation), not to the source-based nonorographic GW scheme.

- Following reviewer's comment, discussion about Richter et al. (2010) is deleted (see lines 18–19 on page 3 of the track-change version of the revised manuscript).

References

Richter, J. H., Sassi, F., and Garcia, R. R.: Toward a physically based gravity wave source parameterization in a general circulation model, J. Atmos. Sci., 67, 136–156, https://doi.org/10.1175/2009JAS3112.1, 2010.

3. Section 3. Why do the authors use both ERA-Interim and MERRA fields, if they basically cover the same altitude range? Why not just one reanalysis?

- We think the overlap of the two reanalysis data can help reduce biases especially in regions where the two reanalyses have quite different structure.
- Each reanalysis has its own reliable altitude range or focuses more on particular altitude range. ERA-Interim reanalysis data are available up to 0.1 hPa for model-level data, but we did not use the ERA-Interim in the mesosphere where effects of spurious Rayleigh damping used instead of nonorographic gravity-wave drag parameterization become large (Fujiwara et al. 2017). In the mesosphere, we used the MERRA2 reanalysis data (together with the NOGAPS-ALPHA) because the Microwave Limb Sounder (MLS) data on the AURA satellite, not used in the ERA-Interim, are assimilated in the MERRA2 (Gelaro et al. 2017). The NOGAPS-ALPHA uses the sounding of the atmosphere using broadband emission radiometry (SABER)

[Figure]

data on the thermosphere ionosphere mesosphere energetics and dynamics (TIMED) satellite in addition to the Aura MLS in the mesosphere (Eckermann et al. 2009).

References

Eckermann, S. D., Hoppel, K. W., Coy, L., McCormack, J. P., Siskind, D. E., Nielsen, K., et al.: High-altitude data assimilation system experiments for the northern summer mesosphere season of 2007, J. Atmos. Solar-Terr. Phys., 71, 531–551, https://doi.org/10.1016/j.jastp.2008.09.036, 2009.
Fujiwara, M., Wright, J. S., Manney, G. L., Gray, L. J., Anstey, J., Birner, T., et al.: Introduction to the SPARC Reanalysis Intercomparison Project (S-RIP) and overview of the reanalysis systems, Atmos. Chem. Phys., 17, 1417–1452, https://doi.org/10.54194/acp-17-1417-2017, 2017.
Gelaro, R., McCarty, W., Suárez, M. J., Todling, R., Molod, A., Takacs, L., et al.: The Modern-Era Retrospective Analysis for Research and Applications, Version 2 (MERRA-2), J. Climate, 30, 5419–5454, https://doi.org/10.1175/JCLI-D-16-0758.1, 2017.

4. Page 10 line 11-12. "Zonal F p s in each OGW ensemble member have locally substantial deviations from the ensemble mean (Fig. 3c) in the major mountain areas". This may be true, but it is not discernible in the figure.

   • Following reviewer's comment, Fig. 3 is modified. Note that some panels are removed following the reviewer 2's comments (see page 35 of the track-change version of the revised manuscript).

5. Figure 8. I may be missing something, but how is it possible that the number of GW packets increase with height in the 2D simulation? If I understand correctly, in the 2D case the only process adding wave packets to a given column is wave generation at the source level.

[Figure]

- As reviewer said, GW packets are regularly launched upward at the source levels, but their upward propagation is not uniform in the vertical direciton even in the 2D simulation. The vertical group velocities ($c_{gz}$) can vary in the vertical direction depending on the background wind and temperature. That is, GW packets can converge (diverge) in the vertical direction when $\partial c_{gz}/\partial z < 0$ ($\partial c_{gz}/\partial z > 0$).
* * *

---

## Author Comment (AC3) · 28 Apr 2020

**General comments**

Overall this is a nice study that puts together a really unique and well-designed set of experiments. The math is explained very clearly and completely and the paper provides a well-documented citation list. Moreover, just getting the various GWD schemes running in WACCM is a noteworthy achievement. In terms of the application of the new GWD packages to a problem, I am excited to see the topic of the effects of GWD on SSWs see more attention and the inclusion

of orographic and nonorographic schemes in a sophisticated model setting like WACCM offers the possibility to explore some noteworthy questions.

That said, I do feel like some of those noteworthy questions were not addressed and I think that this is a real missed opportunity. In the text below, I suggest a few ideas (which would require a few additional figures) that I think would be very worth the effort to include. I suggest these ideas because as it stands, this paper does not really discuss the mechanistic effects of GWD on SSWs, rather it simply provides some momentum budgets. Not that providing momentum budgets is not interesting, I just really think that a few additional figures could turn this paper into a something of very high value to the community. To be clear, my acceptance of this paper is not contingent on the authors adding my suggestions, I am simply trying to help improve the relevance of the paper.

- ■ Authors would like to thank the reviewer for reading and evaluating the original manuscript. We think that the reviewer's questions really help improve discussions on the time evolution of the GW pseudomomentum fluxes in our original manuscript. We have added more figures (Figs. 13–14) and related discussions in our revised manuscript according to reviewer's comments. Please refer to the track-change version of revised manuscript for figure, page and line numbers to be mentioned below.

**Major comments**

1. There are two overarching concepts that I think would make compelling additions to your paper. One involves the effects of GWD on the pre-warming evolution of the vortex (i.e., preconditioning) and the second involves the possibility that GWD increases or decreases the probability of SSW occurrence. For each of these topics, I suggest two figures from the current literature that would provide a good starting point for figures to provide in the current manuscript.

2. Topic one – preconditioning: Your paper only shows figures for Jan. 20, but one could argue that it is the overall vortex evolution from Jan. 10-20 that is of prime interest in understanding the triggering of this particular SSW. Indeed Figs. 6-10 of Albers and Birner (JAS 2014) show that this period was of notable interest in the development of the SSW and in particular for GWD, it is the zonally asymmetric momentum fluxes that may play an important role in SSW development. Thus can you provide a few additional figures that show the differences in the vortex evolution and zonally asymmetric momentum fluxes for Jan. 10-20? In particular, zonally asymmetric views with GW momentum fluxes and geopotential height contours to indicate vortex shape (as in Figs. 6 and 7 of Albers and Birner) would be very interesting for the various cases that you have run.

   • Following the reviewer's suggestion, two new figures (Figs. 13–14) are added in the revised manuscript. Please find the new figures on the pages 45–46 of the track-change version of the revised manuscript. Thanks to this reviewer's question, we realized we missed opportunity to discuss the importance of the enhanced eastward OGW pseudomomentum fluxes in the middle stratosphere in the early stage of the SSW evolution. Newly added discussions can be found from the bottom of the page 18 to the middle of the page 20. Figures 13 and 14 demonstrate that the zonal-wavenumber-2 structure of the OGW pseudomomentum fluxes is much more enhanced in the middle stratosphere on 11 January 2009 in the 4D than in the 2D. Also, we discuss that this enhanced zonal-wavenumber-2 structure in the 4D interact more actively with the polar vortex in the early period, inducing the formation of the polar vortex of the zonal-wavenumber-2 structure (i.e., Rossby waves with the zonal-wavenumber-2 structure).

3. Topic two – probability of SSW occurrence: I'm not sure that my second suggestion is possible to accomplish with your current ensemble setup, but in case it is possible, I think it would provide a very interesting result. In de la Camara

et al. (JAS 2017), it was shown that perturbations to the vortex prior to a SSW can cause the vortex to evolve in very different ways. In particular, Figs. 2b, 6b, and 7 provide a very interesting way of seeing how perturbations to the vortex can disrupt vortex evolution, and in some cases, even disrupting the SSW from occurring at all. Now, I realize that your ensembles start about two weeks before the SSW central date, which means that most (all?) of your ensembles have a SSW, but even so, are there systematic differences in how the vortex evolves for the different model setups? Are there ensembles where a SSW does not occur or just barely occurs? In a similar fashion to what I suggest above, I would be particularly interested in seeing figures similar to de la Camara et al. Fig.7 (which is itself similar in character to Figs. 6 and 7 of Albers and Birner); that is, how does a stereographic view of the vortex evolution look between the various model setups?

- In the present study, we cannot show how sensitive the evolution of polar vortex is to the perturbations in the stratosphere because the simulations are all offline calculations carried out for fixed time evolution of the large-scale flow. We can understand that there is a possibility of active interaction between GWs and polar vortex (or planetary waves) when the 4D formulations are employed. However, we cannot actually measure how much the different spatial distributions of the GW pseudomomentum fluxes can affects the vortex evolution. But, we added some discussions about the sensitivity to the stratospheric flow in summary and discussion (see texts from the line 34 on the page 21 to the line 12 on the page 22 of the track-change version of the revised manuscript.

**Minor comments**

1. Figure 9, 10 and S7: These figures are quite difficult to read. Since you are really

only concerned with the NH, why not truncate the figures to include on the NH, or perhaps even just 30-90 N?

- Following the review's suggestion, the southern hemispheric parts of Figs. 9, 10 and S7 are truncated. Please find the modified Figs 9 and 10 on the pages 41 and 42 of the track-change version of the revised manuscript.
* * *

---

## Author Response (ED1)

**Authors' Responses to Reviewer #1's comments for ACP-2019-1046**

April 28, 2020

**General comments**

This paper investigates the effect of four-dimensional propagation of gravity waves in time-varying background winds on their properties (pseudomomentum fluxes, wavenumber) during the occurrence of a sudden stratospheric warming. The main motivation is that GW parameterizations implemented in climate models generally neglect these effects (columnar, instantaneous propagation is generally enforced in the GW schemes), and it is important to assess the missing effects on the redistribution of momentum flux and GW forcing. The authors do not find a big difference between 4D and 2D propagation in terms of latitude-height structure of the total momentum fluxes, but do find a significant difference in terms of the magnitude of the momentum fluxes, with much larger fluxes in the 4D scheme. The effects of curvature on the magnitude of the fluxes seems to be as important as the effect of horizontal wind shear.

The study is well-written and easy to follow, and the results are relevant and timing, aligning with current efforts to better understand GW processes in order to improve their parameterizations in climate models. I have a few, very minor comments, and I believe the paper is basically publishable as is.

- ■ Authors would like to thank the reviewer for reading and evaluating the original manuscript. We have corrected our manuscript according to the reviewer's comments. Please refer to the track-change version of our revised manuscript for figure and line numbers to be mentioned below.

**Minor comments**

1. Some parts of the introduction seem a succession of references, and sometimes it is difficult to follow/understand the line of argument (e.g., paragraphs in page 2).

   - Following reviewer's comment, some redundant references are excluded and the Introduction is somewhat reduced (see pages 2–3 in the track-change version of the revised manuscript).

2. Page 3, line 8-9. Richter et al (2010) attributed the improvement in the SSW frequency in WACCM to the turbulent mountain stress parameterization (which improves near-surface winds and planetary wave generation), not to the source-based nonorographic GW scheme.

   - Following reviewer's comment, discussion about Richter et al. (2010) is deleted (see lines 18–19 on page 3 of the track-change version of the revised manuscript).

   References

Richter, J. H., Sassi, F., and Garcia, R. R.: Toward a physically based gravity wave source parameterization in a general circulation model, J. Atmos. Sci., 67, 136–156, https://doi.org/10.1175/2009JAS3112.1, 2010.

3. Section 3. Why do the authors use both ERA-Interim and MERRA fields, if they basically cover the same altitude range? Why not just one reanalysis?

- We think the overlap of the two reanalysis data can help reduce biases especially in regions where the two reanalyses have quite different structure.

- Each reanalysis has its own reliable altitude range or focuses more on particular altitude range. ERA-Interim reanalysis data are available up to 0.1 hPa for model-level data, but we did not use the ERA-Interim in the mesosphere where effects of spurious Rayleigh damping used instead of nonorographic gravity-wave drag parameterization become large (Fujiwara et al. 2017). In the mesosphere, we used the MERRA2 reanalysis data (together with the NOGAPS-ALPHA) because the Microwave Limb Sounder (MLS) data on the AURA satellite, not used in the ERA-Interim, are assimilated in the MERRA2 (Gelaro et al. 2017). The NOGAPS-ALPHA uses the sounding of the atmosphere using broadband emission radiometry (SABER) data on the thermosphere ionosphere mesosphere energetics and dynamics (TIMED) satellite in addition to the Aura MLS in the mesosphere (Eckermann et al. 2009).

References

Eckermann, S. D., Hoppel, K. W., Coy, L., McCormack, J. P., Siskind, D. E., Nielsen, K., et al.: High-altitude data assimilation system experiments for the northern summer mesosphere season of 2007, J. Atmos. Solar-Terr. Phys., 71, 531–551, https://doi.org/10.1016/j.jastp.2008.09.036, 2009.

Fujiwara, M., Wright, J. S., Manney, G. L., Gray, L. J., Anstey, J., Birner, T., et al.: Introduction to the SPARC Reanalysis Intercomparison Project (S-RIP) and overview of the reanalysis systems, Atmos. Chem. Phys., 17, 1417–1452, https://doi.org/10.54194/acp-17-1417-2017, 2017.

Gelaro, R., McCarty, W., Suárez, M. J., Todling, R., Molod, A., Takacs, L., et al.: The Modern-Era Retrospective Analysis for Research and Applications, Version 2 (MERRA-2), J. Climate, 30, 5419–5454, https://doi.org/10.1175/JCLI-D-16-0758.1, 2017.

4. Page 10 line 11-12. "Zonal F p s in each OGW ensemble member have locally substantial deviations from the ensemble mean (Fig. 3c) in the major mountain areas". This may be true, but it is not discernible in the figure.

- Following reviewer's comment, Fig. 3 is modified. Note that some panels are removed following the reviewer 2's comments (see page 35 of the track-change version of the revised manuscript).

5. Figure 8. I may be missing something, but how is it possible that the number of GW packets increase with height in the 2D simulation? If I understand correctly, in the 2D case the only

process adding wave packets to a given column is wave generation at the source level.

- As reviewer said, GW packets are regularly launched upward at the source levels, but their upward propagation is not uniform in the vertical direciton even in the 2D simulation. The vertical group velocities ($c_{gz}$) can vary in the vertical direction depending on the background wind and temperature. That is, GW packets can converge (diverge) in the vertical direction when $\partial c_{gz}/\partial z < 0$ ($\partial c_{gz}/\partial z > 0$).

**Authors' Responses to Reviewer #2's comments for ACP-2019-1046**

April 28, 2020

**General comments**

**Recommendation**: Accept with minor revisions

This is a nicely done, comprehensive study of gravity wave (GW) propagation from tropospheric sources into the middle atmosphere under background wind conditions prevailing during the sudden stratospheric warming of 2009. The authors use a ray tracing model to show how spatial inhomogeneity and evolution of the background flow alters the characteristics of propagating GW; and how this 4D ($x$, $y$, $z$; $t$) propagation model differs from 2D ($z$; $t$) propagation in many nontrivial ways.

The study is well-written and easy to follow, and the results are relevant and timing, aligning with current efforts to better understand GW processes in order to improve their parameterizations in climate models. I have a few, very minor comments, and I believe the paper is basically publishable as is.

- ■ Authors would like to thank the reviewer for carefully reading and evaluating the original manuscript. We think we have corrected faithfully our original manuscript according to reviewer's comments. Please refer to the track-change version of revised manuscript for figure, page and line numbers to be mentioned below.

**Specific comments (page, line)**

1. (1, 16) "may have profound impacts": Why "may"? GW are the main component of the eddy momentum budget in the mesosphere and above. I would have written "have profound impacts".

   - "may" is deleted in the line 19 on the page 1 of the track-change version of the revised manuscript.

2. (1, 24) "radiatively-driven latitudinal temperature gradient across the two poles" → "pole-to-pole radiatively-driven latitudinal temperature gradient".

   - The phrase is modified in the line 4 on the page 2 of the track-change version of the revised manuscript following the reviewer's suggestion.

3. (2, 1) "Irreversible heat and heat fluxes": This does not make sense. I believe you mean to say "irreversible heat and momentum flux divergences".

   - Modified sentences can be found in the lines 5–7 on the page 2 of the track-change version of the revised manuscript.

4. (3, 1) "predominance of non-dissipative wave-mean interaction": "Predominance" overstates the case. Kruse and Smith (2018) stated (their abstract) that "Non-dissipative accelerations are non-negligible and influence a [mountain wave's] approach to breaking, *but breaking and dissipative decelerations quickly develop and dominate the subsequent evolution*" (my italics). Perhaps you meant to say "importance" or "relevance" of non-dissipative interactions? In any case, irreversible changes of the background flow ultimately occur only through dissipation.

   - Modified sentences can be found in the lines 6–12 on the page 3 in the track-change version of the revised manuscript.

5. (3, 18) "GW activities" → GW activity (this is the standard usage, "activity" here being used as a collective noun).

   - Modifications can be found in several places on the pages 3 and 21 of the track-change version of the revised manuscript.

6. (4, 19) "where $\Lambda_n$s ($n = 1, \cdots, N$) denote": This is awkward and confusing because the trailing "s", which I believe is intended to denote a plural, could be taken to be part of the symbol. The standard usage for mathematical symbols is that they do not normally takes an "s" to denote plural. Replace this with "where $\Lambda_n$ ($n = 1, \cdots, N$) denote ...". Note that this occurs many other times through the paper when referring to $\Lambda_n$ and other symbols. Please do a thorough check.

   - $\Lambda_n$s is changed into $\Lambda_n$ in the line 30 on the page 4 of the track-change version of the revised manuscript.

7. (7, 19) "Then, $\tau_{\text{def}}$s are computed" → Then, $\tau_{\text{def}}$ is computed. See previous comment. In standard usage $\tau_{\text{def}}$ stands for all cases of the "deformation" time scale. No trailing "s" needed.

   - The trailing "s" is removed in the line 28 on the page 7 of the track-change version of the revised manuscript.

8. (10, 4) Figure 3: I would delete panels (a) and (d) of this figure, which do not contain any information that cannot be succinctly explained in the text. On the other hand, there could be a little more discussion of the interesting panels (b)-(c) and (e)-(f). In particular, panel (b) indicates that OGW flux, $F_p$, is well organized in space in a single ensemble member. I presume this is due to the fact that $F_p$ is strongly constrained by the OGW source parameterization, which depends explicitly on orography and low-level wind. By contrast, organization of $F_p$ for the NOGW case only emerges in the ensemble because any single ensemble member is completely stochastic (panels (e) vs. (f)).

   - Following the reviewer's suggestion, Fig. 3 is modified, and thus panels for the stochastic parameters are removed (see the page 35 of the track-change version of the revised manuscript). Also, more discussions are added at the end of the page 10 and at the beginning of the page 11 of the track-change version of the revised manuscript.

9. (10, 32) "but being weakened" $\rightarrow$ but is much weakened.

- The phase is changed as suggested in the line 17 on the page 11 of the track-change version of the revised manuscript.

10. (10, 32) "Transparently shaded areas": This is confusing. "Transparent" implies no shading at all. I believe you are referring to the areas overlain by gray(ish) shading. If so, please explain more clearly. Better yet would be to use some other means (e.g., cross-hatching) to denote the regions of non-significant differences to avoid confusion with the color shading meant to denote flux magnitude/sign.

- Following the reviewer's suggestion, the shaded areas are replaced with hatched areas in all the figures that contained the "transparently shaded regions". Also, "transparently shaded" is changed into "hatched" everywhere in the main text of the revised manuscript.

11. (11, 4): "westward Fps in the 4D are about 10 (28) times enhanced ...": This sentence is nearly incomprehensible. Please break it up into two digestible parts, the first referring to the 10X difference between 4D and 2D models in all but one of the parameterizations; and the second referring to the 28X difference in the case of the WM96b non-orographic parameterization. Also, omit the "s" at the end of $F_p$, here and in many other instances; see comment (4, 19).

- The sentence is broken into two around the lines 21–24 on the page 11 of the track-change version of the revised manuscript.

12. (11, 26) "zonal-mean ks": Here and elsewhere this should be "zonal-mean k"; see comment (4, 19).

- $k$s is replaced with $k$ everywhere as well as in the line 13 on the page 12 of the track-change version of the revised manuscript.

13. (12, 6) "thermodynamic forcing terms": What are these? Are you referring to the dependence on N?

- "thermodynamic forcing terms" is removed, and some explanations are added in the line 28–29 on the page 12 of the track-change version of the revised manuscript.

14. (12, 29) "meridional wavenumbers (ls)" $\rightarrow$ Here and elsewhere, "ls" should be replaced simply by "l"; see comment (4, 19).

- "$l$s" is replaced by "$l$" in the line 18 on the page 13 of the track-change version of the revised manuscript. Plural from of mathematical symbols is modified everywhere in the revised manuscript following the reviewer's comment.

15. (13, 19) Figure 8: This figure shows the striking difference between the 4D and 2D models, especially in the discontinuous (in latitude) appearance of OGW $F_p$. This is a common problem in comprehensive global models, which usually employ 2D columnar GW parameterizations. Although wave-mean flow interaction will tend to reduce these effects, this does not necessarily happen for the right reasons; see discussion about "compensation" of parameterized vs. resolved wave forcing in Cohen et al. (JAS 2013, 2014). It might be worth mentioning this problem.

   - Following the reviewer's comments, some discussions are added in the lines 9–20 on the page 15 and the lines 12–14 on the page 21 of the track-change version of the revised manuscript.

16. (15, 19) Figure 11: You might consider showing panels (a) through (f) in vector form (vector background wind, $\boldsymbol{U}$, vector horizontal group velocity, $\boldsymbol{c}_g$). This would show more clearly the relationship between $\boldsymbol{U}$ and $\boldsymbol{c}_g$; and also (for the intrinsic group velocity) the regions where that vector is non-negligible.

   - Following the reviewer's comments, some panels on Fig. 11 are replotted in vector field format. Please see the page 43 of the track-change version of the revised manuscript. Related discussions are rewritten for clarification on the page 17 of the track-change version of the revised manuscript.

17. (17, 12) "These enhanced eastward $F_p$s, if they exist, may induce more rapid recovery of the stratospheric jets, accelerating downward movement of the ES": This is an interesting effect, which would not be captured by the 2D columnar parameterizations used in most comprehensive models. Note again that "$F_p$s" should be simply "$F_p$" (no trailing "s").

   - Please see the lines 28–29 on the page 18 of the track-change version of the revised manuscript. Again, the plural form of mathematical sysbols is modified everywhere in the revised manuscript as the reviewer suggested.

**Authors' Responses to Reviewer #3's comments for ACP-2019-1046**

April 28, 2020

**General comments**

Overall this is a nice study that puts together a really unique and well-designed set of experiments. The math is explained very clearly and completely and the paper provides a well-documented citation list. Moreover, just getting the various GWD schemes running in WACCM is a noteworthy achievement. In terms of the application of the new GWD packages to a problem, I am excited to see the topic of the effects of GWD on SSWs see more attention and the inclusion of orographic and nonorographic schemes in a sophisticated model setting like WACCM offers the possibility to explore some noteworthy questions.

That said, I do feel like some of those noteworthy questions were not addressed and I think that this is a real missed opportunity. In the text below, I suggest a few ideas (which would require a few additional figures) that I think would be very worth the effort to include. I suggest these ideas because as it stands, this paper does not really discuss the mechanistic effects of GWD on SSWs, rather it simply provides some momentum budgets. Not that providing momentum budgets is not interesting, I just really think that a few additional figures could turn this paper into a something of very high value to the community. To be clear, my acceptance of this paper is not contingent on the authors adding my suggestions, I am simply trying to help improve the relevance of the paper.

- ■ Authors would like to thank the reviewer for reading and evaluating the original manuscript. We think that the reviewer's questions really help improve discussions on the time evolution of the GW pseudomomentum fluxes in our original manuscript. We have added more figures (Figs. 13–14) and related discussions in our revised manuscript according to reviewer's comments. Please refer to the track-change version of revised manuscript for figure, page and line numbers to be mentioned below.

**Major comments**

1. There are two overarching concepts that I think would make compelling additions to your paper. One involves the effects of GWD on the pre-warming evolution of the vortex (i.e., preconditioning) and the second involves the possibility that GWD increases or decreases the probability of SSW occurrence. For each of these topics, I suggest two figures from the current literature that would provide a good starting point for figures to provide in the current manuscript.

2. Topic one – preconditioning: Your paper only shows figures for Jan. 20, but one could argue that it is the overall vortex evolution from Jan. 10-20 that is of prime interest in understanding the triggering of this particular SSW. Indeed Figs. 6-10 of Albers and Birner (JAS 2014) show that this period was of notable interest in the development of the SSW and in particular for GWD, it is the zonally asymmetric momentum fluxes that may play an important role in

SSW development. Thus can you provide a few additional figures that show the differences in the vortex evolution and zonally asymmetric momentum fluxes for Jan. 10-20? In particular, zonally asymmetric views with GW momentum fluxes and geopotential height contours to indicate vortex shape (as in Figs. 6 and 7 of Albers and Birner) would be very interesting for the various cases that you have run.

- Following the reviewer's suggestion, two new figures (Figs. 13–14) are added in the revised manuscript. Please find the new figures on the pages 45–46 of the track-change version of the revised manuscript. Thanks to this reviewer's question, we realized we missed opportunity to discuss the importance of the enhanced eastward OGW pseudo-momentum fluxes in the middle stratosphere in the early stage of the SSW evolution. Newly added discussions can be found from the bottom of the page 18 to the middle of the page 20. Figures 13 and 14 demonstrate that the zonal-wavenumber-2 structure of the OGW pseudomomentum fluxes is much more enhanced in the middle strato-sphere on 11 January 2009 in the 4D than in the 2D. Also, we discuss that this enhanced zonal-wavenumber-2 structure in the 4D interact more actively with the polar vortex in the early period, inducing the formation of the polar vortex of the zonal-wavenumber-2 structure (i.e., Rossby waves with the zonal-wavenumber-2 structure).

3. Topic two – probability of SSW occurrence: I'm not sure that my second suggestion is possi-ble to accomplish with your current ensemble setup, but in case it is possible, I think it would provide a very interesting result. In de la Camara et al. (JAS 2017), it was shown that pertur-bations to the vortex prior to a SSW can cause the vortex to evolve in very different ways. In particular, Figs. 2b, 6b, and 7 provide a very interesting way of seeing how perturbations to the vortex can disrupt vortex evolution, and in some cases, even disrupting the SSW from occurring at all. Now, I realize that your ensembles start about two weeks before the SSW central date, which means that most (all?) of your ensembles have a SSW, but even so, are there systematic differences in how the vortex evolves for the different model setups? Are there ensembles where a SSW does not occur or just barely occurs? In a similar fashion to what I suggest above, I would be particularly interested in seeing figures similar to de la Ca-mara et al. Fig.7 (which is itself similar in character to Figs. 6 and 7 of Albers and Birner); that is, how does a stereographic view of the vortex evolution look between the various model setups?

- In the present study, we cannot show how sensitive the evolution of polar vortex is to the perturbations in the stratosphere because the simulations are all offline calculations carried out for fixed time evolution of the large-scale flow. We can understand that there is a possibility of active interaction between GWs and polar vortex (or planetary waves) when the 4D formulations are employed. However, we cannot actually measure how much the different spatial distributions of the GW pseudomomentum fluxes can affects the vortex evolution. But, we added some discussions about the sensitivity to the strato-spheric flow in summary and discussion (see texts from the line 34 on the page 21 to the line 12 on the page 22 of the track-change version of the revised manuscript.

**Minor comments**

1. Figure 9, 10 and S7: These figures are quite difficult to read. Since you are really only concerned with the NH, why not truncate the figures to include on the NH, or perhaps even just 30-90 N?

   - Following the review's suggestion, the southern hemispheric parts of Figs. 9, 10 and S7 are truncated. Please find the modified Figs 9 and 10 on the pages 41 and 42 of the track-change version of the revised manuscript.

[revised manuscript text omitted]

OGWs at z = 38.0 km at 00UTC on 20 January 2009

(a) **U** → 100 m s⁻¹

(b) **c**$_{gh}$ (4D) → 100 m s⁻¹

(c) **c**$_{gh}$ − **U** (4D) → 10 m s⁻¹

(d) c$_{gz}$ (4D)

(e) ω̂/|f| (4D)

**Figure 11.** Longitude-latitude cross-sections of (a)  horizontal wind ($\cancel{U}$$U$),  ensemble means of  ( d b) the horizontal group velocity ($\cancel{c_{g\lambda}\ and\ c_{g\phi}}$$c_{gh}$) , ( f c) the horizontal intrinsic group velocity ($\cancel{c_{g\lambda}-U\ and\ c_{g\phi}-V}$$c_{gh}-U$), ( d) the vertical component of the group velocity ($c_{gz}$), and ( e) ratio of intrinsic frequency ($\hat{\omega}$) to Coriolis parameter ($|f|$) for  OGW at $z = 38$ km in the 4D experiment at 00 UTC on 20 January, 2009.  Hatched areas indicate regions where the paired and two-tailed $t$-test for the 4D and 2D experiments gives $p$ values larger than 0.05 (i.e., no statistical significance at the level of 0.05). For hatching over horizontal vector fields (b–c), the mean value of $p$ values for the zonal and meridional components is used.

[Figure]

**Figure 12.** Time-height cross-sections of (a) zonal wind ($U$), and (b–e) ensemble means of zonal pseudomomentum fluxes ($F_p$s) averaged over 30°N–90°N for  OGW and NOGW$_{SA11}$  in the (top) 4D and (bottom) 2D experiments. OGW $F_p$  is multiplied by the efficiency factor (0.125). Contour interval for zonal winds is 10 m s$^{-1}$ and negative values are plotted in dashed lines.  Hatched areas over the zonal $F_p$  indicate regions where the paired and two-tailed $t$-test for the 4D and 2D experiments gives $p$ values larger than 0.05 (i.e., no statistical significance at the level of 0.05).

**Relative vorticity at 5 hPa and zonal $F_p$ of OGWs at z = 38 km**

[Figure]

**Figure 13.** Relative vorticity at 5 hPa and zonal pseudomomentum fluxes ($F_p$) for OGW at $z = 38$ km at 00 UTC on (a, e) 11 January, (b, f) 15 January, (c, g) 17 January, and (d, h) 19 January in 2009 for the (top) 4D and (bottom) 2D experiments. OGW $F_p$ is multiplied by the efficiency factor (0.125). Contour interval for relative vorticity is $5 \times 10^{-5}$ s$^{-1}$ and negative values are plotted in dashed lines. Hatched areas over the zonal $F_p$ indicate regions where the paired and two-tailed $t$-test for the 4D and 2D experiments gives $p$ values larger than 0.05 (i.e., no statistical significance at the level of 0.05). Latitudinal grids are plotted every $10°$ from $20°$N

Relative vorticity at 5 hPa and meridional F_p of OGWs at z = 38 km

[Figure]

**Figure 14.** Same as Fig. 13 except for meridional pseudomomentum fluxes ($F_p$).